# Sialoglycan binding triggers spike opening in a human coronavirus

Matti F. Pronker[1], Robert Creutznacher[2], Ieva Drulyte[3], Ruben J. G. Hulswit[2], Zeshi Li[4], Frank J. M. van Kuppeveld[2], Joost Snijder[1], Yifei Lang[2,6], Berend-Jan Bosch[2], Geert-Jan Boons[4], Martin Frank[5], Raoul J. de Groot[2 ✉] & Daniel L. Hurdiss[2 ✉]

Coronavirus spike proteins mediate receptor binding and membrane fusion, making them prime targets for neutralizing antibodies. In the cases of severe acute respiratory syndrome coronavirus, severe acute respiratory syndrome coronavirus 2 and Middle East respiratory syndrome coronavirus, spike proteins transition freely between open and closed conformations to balance host cell attachment and immune evasion[1–5]. Spike opening exposes domain S1[B], allowing it to bind to proteinaceous receptors[6,7], and is also thought to enable protein refolding during membrane fusion[4,5]. However, with a single exception, the pre-fusion spike proteins of all other coronaviruses studied so far have been observed exclusively in the closed state. This raises the possibility of regulation, with spike proteins more commonly transitioning to open states in response to specific cues, rather than spontaneously. Here, using cryogenic electron microscopy and molecular dynamics simulations, we show that the spike protein of the common cold human coronavirus HKU1 undergoes local and long-range conformational changes after binding a sialoglycan-based primary receptor to domain S1[A]. This binding triggers the transition of S1[B] domains to the open state through allosteric interdomain crosstalk. Our findings provide detailed insight into coronavirus attachment, with possibilities of dual receptor usage and priming of entry as a means of immune escape.

Long before the advent of severe acute respiratory syndrome coronavirus 2 (SARS-CoV-2), four coronaviruses (CoVs) colonized the human population. Two of these, human CoVs HKU1 and OC43 in the betacoronavirus subgenus *Embecovirus*, independently arose from rodent reservoirs—either directly or through intermediate hosts[8–10]. Unlike other human CoVs, HKU1 and OC43 rely on cell surface glycans as indispensable primary receptors[11,12]. Their attachment and fusion spike proteins specifically bind to 9-*O*-acetylated sialosides[11,13–17]. Underlining the importance of glycan attachment, embecoviruses uniquely code for an additional envelope protein, haemagglutinin esterase, a sialate-*O*-acetylesterase serving as a receptor-destroying enzyme[13,18,19]. Recent observations suggest that HKU1 spike particularly targets α2,8-linked 9-*O*-acetylated disialosides (9-*O*-Ac-Sia(α2,8)Sia; that is, glycan motifs typical of oligosialogangliosides such as GD3). Accordingly, following overexpression of GD3 synthase ST8SIA1, HEK293T cells become susceptible to HKU1 S-pseudotyped viruses[17].

CoV spike proteins are homotrimeric class I fusion proteins[20]. The spike protomer can be divided into an amino- and carboxy-terminal region designated S1 and S2, respectively. Distinct S1 domains mediate receptor binding[21], whereas S2 comprises the fusion machinery (Fig. 1a). In HKU1 and OC43, attachment to 9-*O*-Ac-sialosides occurs through a well-conserved receptor-binding site located in spike protein domain S1[A] (Fig. 1a)[15,16]. There are indications, however, for the existence of a secondary receptor engaged through domain S1[B], as epitopes of virus-neutralizing antibodies map to subdomain S1[B2] (refs. 22–24). Moreover, in the case of HKU1, recombinantly expressed S1[B] blocks infection[23], with single-site substitutions in S1[B2] abolishing this activity[24].

The spike proteins of SARS-CoV, SARS-CoV-2 and Middle East respiratory syndrome coronavirus (MERS-CoV) occur in different conformations with their receptor-binding S1[B] domains either partially buried between neighbouring protomers ('closed' or 'down') or with one or more S1[B] domains exposed (1-, 2- and 3-up, 'open')[2,5,7,25]. The conformational dynamics of S1[B], and modulation thereof, would provide CoVs with a means to balance host cell attachment and immune escape[1]. Recently, spontaneous conversion of S1[B] into the up conformation was also described for porcine epidemic diarrhoea virus[26]. However, available structures of all other CoV spike proteins, including those of HKU1 and OC43 (refs. 16,27), have been observed only in a closed conformation (Supplementary Table 1), shielding S1[B] from neutralizing antibodies but preventing S1[B]-mediated receptor engagement[1,22]. Adding to the conundrum, the transition from a closed to an open spike conformation has been linked to the elaborate conformational changes in S2 that drive fusion[4,28,29]. The question thus arises whether specific

[1]Biomolecular Mass Spectrometry and Proteomics, Bijvoet Center for Biomolecular Research, Department of Chemistry, Faculty of Science, Utrecht University, Utrecht, The Netherlands. [2]Virology Section, Infectious Diseases and Immunology Division, Department of Biomolecular Health Sciences, Faculty of Veterinary Medicine, Utrecht University, Utrecht, The Netherlands. [3]Materials and Structural Analysis, Thermo Fisher Scientific, Eindhoven, The Netherlands. [4]Department of Chemical Biology and Drug Discovery, Utrecht Institute for Pharmaceutical Sciences, Utrecht University, Utrecht, The Netherlands. [5]Biognos AB, Gothenburg, Sweden. [6]Present address: Research Center for Swine Diseases, College of Veterinary Medicine, Sichuan Agricultural University, Chengdu, China. ✉e-mail: r.j.degroot@uu.nl; d.l.hurdiss@uu.nl

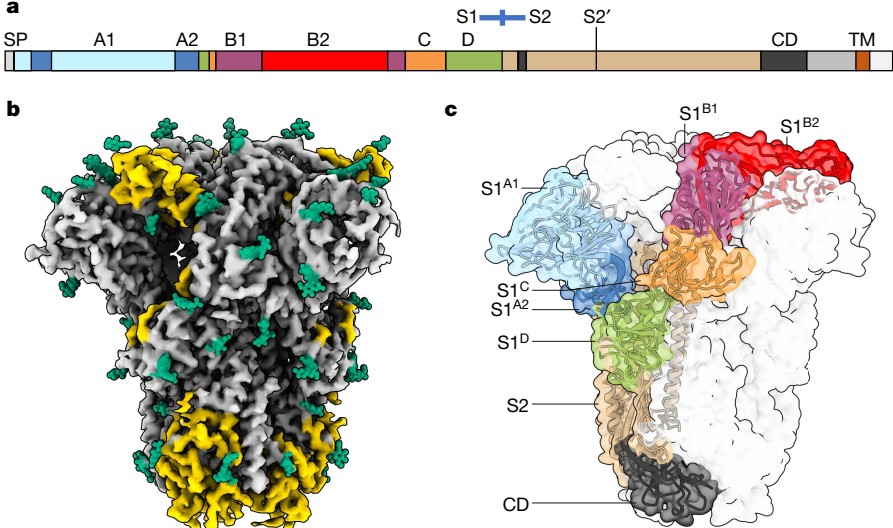

**Fig. 1 | Cryo-EM structure of apo HKU1-A spike protein. a**, A linear representation of the HKU1-A spike protein primary sequence, coloured by domain and with the S1–S2 domains, S2′ protease cleavage site, signal peptide (SP), connecting domain (CD) and transmembrane helix (TM) indicated. **b**, A cryo-EM density map for apo HKU1-A spike protein, with previously unmodelled glycans indicated in green and newly modelled amino acids in yellow. **c**, The apo HKU1-A spike trimer with one Y-shaped protomer coloured by (sub)domain as in **a**.

mechanisms might exist that trigger S1$^B$ conversion to the open state. Here we describe cryogenic electron microscopy (cryo-EM) structures of a serotype A HKU1 (HKU1-A) spike protein in four conformations, one in a closed apo state, the others in complex with the HKU1 disialoside receptor 9-*O*-Ac-Sia(α2,8)Sia. We show that glycan receptor binding by S1$^A$ specifically prompts a conformational transition of S1$^B$ domains into 1- and eventually 3-up positions, apparently through an allosteric mechanism.

## Structure of the apo HKU1-A spike protein

HKU1 field strains are divided into three genotypes with evidence of intertypic recombination, but essentially occur in two distinct serotypes, with either A- or B-type spike proteins[30]. Single-particle cryo-EM analysis of spike protein ectodomains of HKU1-A strain Caen1 yielded a reconstruction for the unbound state at a global resolution of 3.4 Å (Fig. 1b, Extended Data Fig. 1, Supplementary Figs. 1 and 2 and Supplementary Table 2). Notably, the HKU1-A spike protein trimers were found exclusively in a closed, pre-fusion conformation as reported for a serotype B HKU1 (HKU1-B) spike protein[27]. The HKU1-A and HKU1-B spike proteins, at 84% sequence identity (Supplementary Fig. 3), are highly similar in global structure with an average Cα root mean square deviation of 1.1 Å for pruned atom pairs (Extended Data Fig. 2a). Compared to the HKU1-B model, our data allowed building an additional 231 residues per protomer. Among newly built segments are the membrane-proximal connecting domain (residues 776–796 and 1152–1225) and the linker between the S1/S2 and S2′ protease cleavage sites (residues 878–907; Fig. 1c). We could also model a major portion of S1$^{B2}$ (residues 480–575) such that this subdomain—purportedly crucial for protein receptor binding—is now fully resolved in the context of an intact HKU1 spike trimer, our findings essentially confirming the crystal structure of a HKU1-A S1$^{B-C}$ fragment (residues 310–677)[24] (Extended Data Fig. 2b). In addition, 20 N-linked glycans per protomer were built, all well supported by the density map (Fig. 1b). Several glycans are engaged in interprotomer contacts (for example, N1215; Supplementary Fig. 4), among which the S1$^B$ N355-glycan may help stabilize the HKU1-A spike trimer in the closed conformation by contacting the clockwise neighbouring protomer via Y528 (Supplementary Fig. 5). Using site-specific glycosylation patterns of HKU1-B (ref. 31), we carried

out molecular dynamics simulations of the fully glycosylated spike ectodomain trimer. HKU1-A spike is largely shielded by glycans leaving only a few regions exposed, most notably the sialic acid-binding site in domain S1$^A$ (Extended Data Fig. 2c).

Predictably similar in overall arrangement, the apo structures of A- and B-type spike trimers differ in the orientation of their S1$^A$ domains, with those of HKU1-A tilted outwards (Extended Data Fig. 2a). The S1$^A$ 9-*O*-Ac-Sia-binding site is conserved in HKU1-A S1$^A$, as expected, with key ligand contact residues K80, T/S82 and W89 (ref. 15) aligning with those in HKU1-B spike (Extended Data Fig. 2d,e). There are, however, notable differences in binding site topology. In HKU1-B, the 9-*O*-Ac-Sia-binding site is located within a narrow crevice between loop elements e1 (residues 29–37) and e2 (residues 246–252)[15,16]. In the HKU1-A spike apo structure, the p1 and p2 pockets that accommodate the sialoside 9-*O*-Ac and 5-*N*-Ac moieties, respectively, are much less prominent owing to a consequential outward displacement of the e1 loop (see below).

## Glycan binding triggers opening of S1$^B$

Incubation of the HKU1-A spike protein with the receptor analogue 9-*O*-Ac-Neu5Ac-α2,8-Neu5Ac-Lc-biotin (Supplementary Fig. 6) led to marked conformational changes yielding a surprising heterogeneity in structures. We identified and modelled three distinct conformations: a fully closed state (3.8 Å resolution), a partially opened state with a single S1$^B$ domain rotated upwards by 101° (1-up, 5 Å resolution) and a fully opened state (3-up, 3.7 Å resolution; Fig. 2, Extended Data Fig. 3, Supplementary Figs. 2, 7 and 8 and Supplementary Table 2). A 2-up state was not detected. In all holo structures, clear densities for the disialoside were observed within S1$^A$ receptor-binding sites (Fig. 2 and Supplementary Figs. 8 and 9). Apparently, binding of a specific 9-*O*-Ac-Sia-based primary receptor analogue by the S1$^A$ domain triggers an allosteric mechanism, causing the exposure of S1$^B$ domains located 40 Å from the S1$^A$ binding pocket (Fig. 2 and Supplementary Fig. 10).

We propose a stepwise model for ligand-induced spike opening (Supplementary Video 1). In the starting apo state, each S1$^B$ domain is held in place, wedged between the S1$^A$ and S1$^B$ domains of the anticlockwise neighbouring Y-shaped protomer. Of the two observed protein–protein interfaces, the one with S1$^A$ buries a larger surface area (Supplementary Fig. 5; 1,207 Å$^2$ versus 442 Å$^2$). In the presence of the S1$^A$ ligand, most

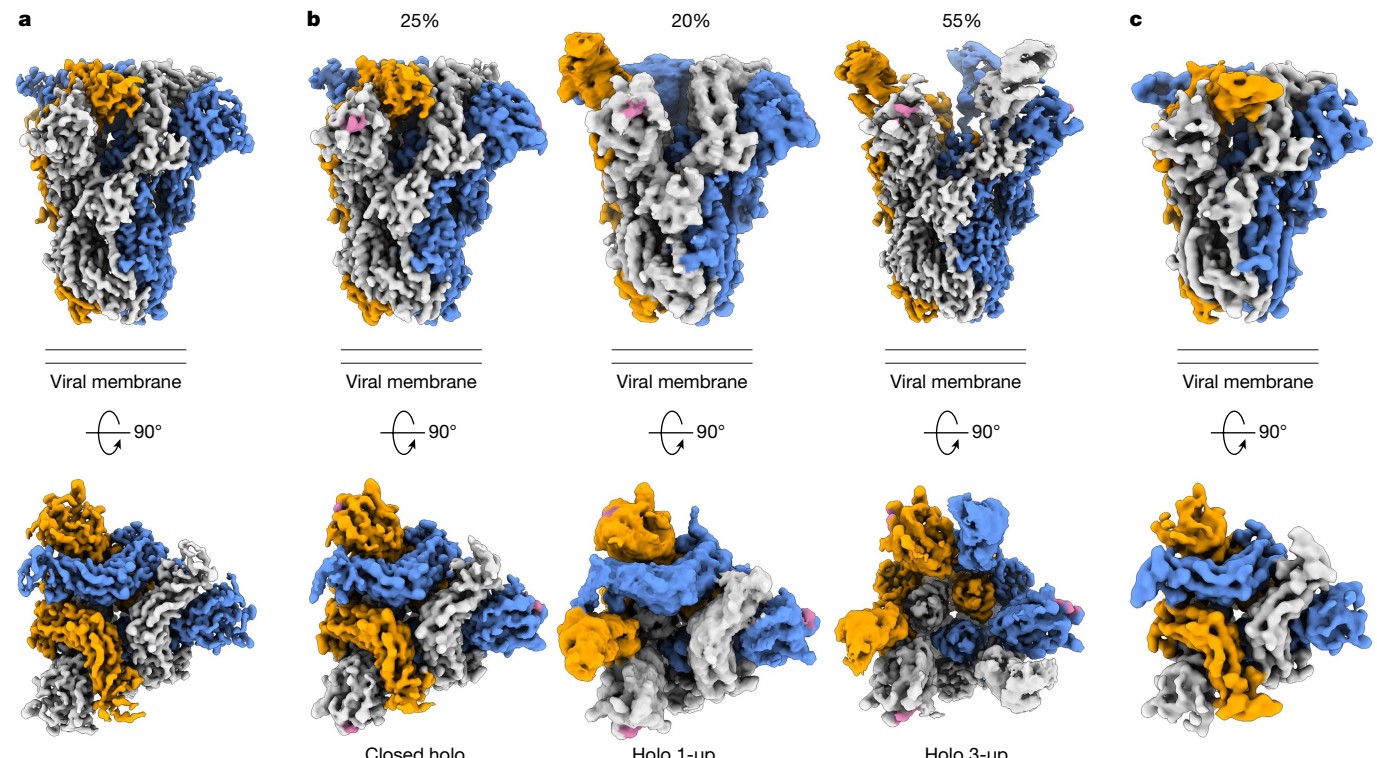

**Fig. 2 | Cryo-EM density maps of wild-type apo HKU1-A spike protein, its complex with a 9-*O*-acetylated disialoside and an equivalently ligand-bound W89A mutant. a**, Orthogonal views of the apo HKU1-A spike trimer density map with protomers coloured in grey, orange and blue. **b**, Density maps of HKU1-A spike protein in complex with the disialoside (in pink). Three distinct classes were observed, with either no, one or three S1[B] domains in the open conformation. Relative occurrence (%) is indicated. **c**, Cryo-EM density map of the W89A mutant HKU1-A spike protein obtained in presence of the disialoside.

spike trimers transitioned into the 1- or 3-up open states. However, 25% of ligand-bound particles remained fully closed. The structure of this 'closed holo' trimer is distinct from that of non-complexed apo trimers, marking it as an initial step in a series of conformational transitions. Ligand binding in the 'closed holo' state is associated with intradomain conformational changes within S1[A]. In particular, the upper S1[A1] subdomain (residues 14–39 and 72–260) rotates inwards by 9° relative to S1[A2] and the remainder of the spike monomer (Fig. 3). Whereas this motion leaves the S1[B]–S1[B] interface unaltered, it has a profound effect on the S1[A]–S1[B] contact area, displacing interfacing residues by approximately 8 Å (Extended Data Fig. 4). This reshaping of the S1[A]–S1[B] interface seems to be the key phenomenon from which subsequent upward rotation of the first S1[B] domain follows, involving a 101° rotation and raising the tip of the S1[B2] subdomain by 50 Å (Fig. 3b, Supplementary Video 2 and Supplementary Fig. 11).

The large conformational change of S1[B] going from the closed holo to the holo up state is accompanied by additional domain rotations of S1[A], S1[C] and S1[D] (Fig. 3b). Conversion into the '1-up' state eliminates the S1[B]–S1[B] interdomain contact. The apparent absence of particles in a '2-up' conformation might be explained by the fact that a lone downward-oriented S1[B] lacks any such stabilizing interactions with neighbouring S1[B] domains, probably making this a transient intermediate.

To rule out the possibility that a subset of open S1[B] domains exist within the apo dataset, we symmetry-expanded the particles from the apo reconstruction and carried out three-dimensional variability analysis on the masked S1[B] domain. No open S1[B] domains in the apo dataset were identified. When the same analysis was carried out on the '1-up' particles, open and closed domains could be easily discriminated, confirming the validity of this approach (Extended Data Fig. 5).

To substantiate our observations, we acquired a dataset with a sialoglycan-binding-defective mutant W89A HKU1 spike[15] in the presence of the 9-*O*-Ac-disialoside as a negative control (Fig. 2c, Extended Data Fig. 6, Supplementary Figs. 2 and 12 and Supplementary Table 2). Again, the spike trimers were all fully closed and morphologically indistinguishable from the unbound apo state of the parental spike protein,

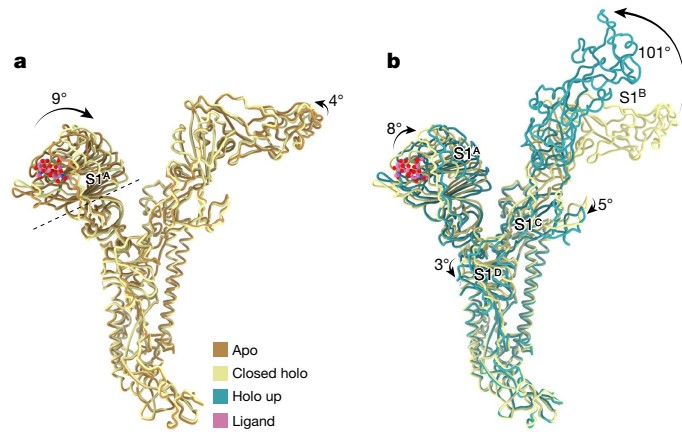

**Fig. 3 | Allosteric interdomain and intradomain rotations are observed following ligand binding. a**, Superposition of single HKU1-A spike protomers in the apo and ligand-bound closed holo state. The intrasubdomain axis of rotation in S1[A] is indicated as a dashed line; the disialoside is shown as spheres with carbon atoms in pink. **b**, Domain rotations associated with transition from the closed holo state to the holo up conformation (3-up state shown).

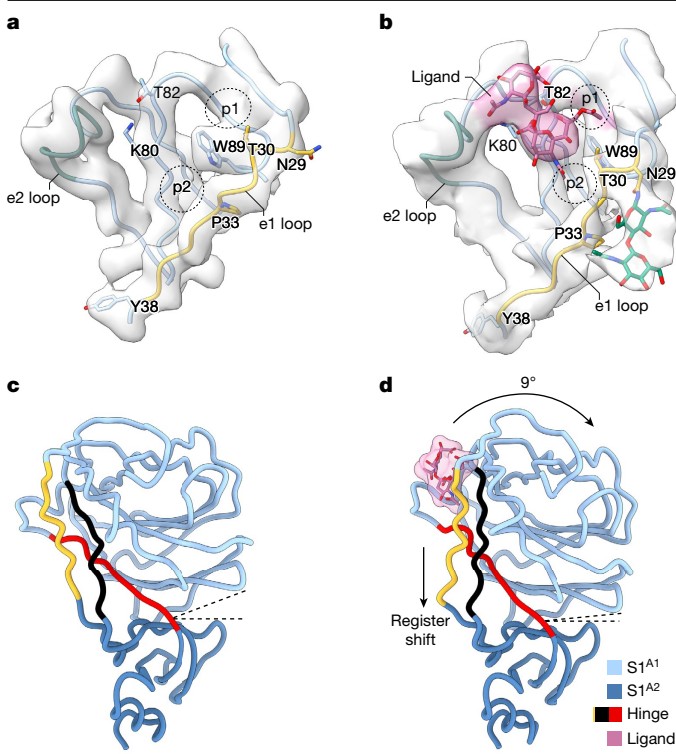

**Fig. 4 | Comparison of the sialic acid-binding site in the apo and closed holo S1$^A$ domains. a,b,** The S1$^A$ sialic acid-binding pocket in the apo (**a**) and holo (**b**) states. The e1 loop (yellow), e2 loop (green) and pockets p1 and p2 are indicated. The disialoside ('ligand') is shown in pink; GlcNAc residues of the N29 N-linked glycan are shown in green. Note the conformational differences in the e1 loop in the holo as compared to the apo state. **c,d,** Side-by-side comparison of the S1$^A$ domain in the apo (**c**) and closed holo state (**d**). The hinge segments connecting subdomains S1$^{A1}$ and S1$^{A2}$ are highlighted with the e1 loop in yellow. Dashed lines indicate the angle between the subdomains.

reinforcing the notion that binding of 9-*O*-Ac-Sia(α2,8)Sia is key for allosteric release of S1$^B$.

## Local conformational changes in S1$^A$

Local refinement of the symmetrical closed structure of the HKU1–ligand complex allowed us to visualize the disialoside bound in the S1$^A$ receptor-binding site (Fig. 4, Supplementary Fig. 8 and Supplementary Table 2), with both Sia moieties discernible. The location of the essential terminal Sia (Sia2) is as expected for a canonical 9-*O*-Ac-Sia-binding site[15] and matches that of the holo cryo-EM structure of OC43 spike protein[16] (Supplementary Fig. 13). Its assigned orientation positions the sialate-9-*O*-acetyl and sialate-5-*N*-acetyl moieties so that they can dock into pockets p1 and p2, respectively, astride the perpendicularly placed W89 side chain. The Sia2 carboxylate is poised to interact with K80 and T82 through a salt bridge and hydrogen bond (Fig. 4b). Using dedicated molecular dynamics simulations of the free disialoside, we identified favourable glycan conformers to restrain modelling of the flexible α2,8-glycosidic linkage and were able to build the outward-facing, reducing-end Sia (Sia1) close to the e2 loop (Extended Data Fig. 7).

Binding of the ligand to the S1$^A$ binding site is accompanied by local conformational changes, most conspicuously involving the displacement of the flanking e1 loop by 3 Å. W89 and T30 are brought in proximity to allow side-chain hydrogen bonding, stabilizing the p1 pocket, and P33 shifts towards the p2 pocket. Concomitantly, the N29 glycan, unresolved in the apo structure, becomes partially ordered

and is displaced by 5 Å away from the S1$^A$–S1$^B$ interface (Fig. 4a,b and Supplementary Video 3). With the N terminus stapled to the S1$^{A1}$ core by means of a disulfide bond (C20–C156), the local changes in e1 are distally translated into long-range conformational changes. These extend all the way down to Y38, some 25 Å away from the binding pocket (Fig. 4a,b and Supplementary Video 4), located within a triple-strand hinge region that links the S1$^{A1}$ and S1$^{A2}$ subdomains (Fig. 4c,d). The resulting register shift between the e1 segment (residues 29–37) and its neighbouring interacting partner (residues 73–81; indicated in black in Fig. 4c,d) seemingly drives the inward 9° rotation of the S1$^{A1}$ subdomain about the S1$^{A1/A2}$ axis (Fig. 4d and Supplementary Video 5).

## MD analysis of S1$^A$

The inherent flexibility of the disialoside-binding pocket limits local resolution and the analysis of inter-residue interactions in our cryo-EM models. To gather atomistic insight into ligand binding, especially of Sia1, and the resulting shift in the protein conformational equilibrium, we carried out molecular dynamics simulations of the S1$^A$ domain on an accumulated timescale of 70 μs.

Simulations starting from the ligand-bound cryo-EM holo structure revealed one dominating disialoside conformer in which the carboxylate of Sia1 interacts through a salt bridge with K84 and the Sia1 5-*N* is stabilized by a hydrogen bond with T82 (Fig. 5a, Extended Data Figs. 8 and 9 and Supplementary Video 6).

Taking an unbiased molecular dynamics approach to the conformational transition of e1, we used our structure of the apo S1$^A$ domain as a starting model. The disialoside was placed into the binding pocket guided by the well-established orientation of 9-*O*-Ac-Sia2. Both the e1 and e2 loops showed pronounced dynamics in all trajectories as shown by a per-residue root mean square deviation analysis (Extended Data Fig. 10a). Saliently, conformational transitions observed in the e1 loop mirrored those identified on comparison of the apo and closed holo cryo-EM models, even though the molecular dynamics data were obtained fully independently (Fig. 5b and Supplementary Video 7). The observations were extended and corroborated by simulations with the S1$^A$ domain of the HKU1-A N1 reference strain[30], which differs from the Caen1 variant in that it carries a tyrosine instead of a lysine at position 84 (Extended Data Fig. 10b). All local conformational changes were observed, although a loss in stabilizing interactions of Sia1 was noted, as would be expected owing to the absence of K84 (Extended Data Fig. 8d and Supplementary Tables 3–5).

In the p1 pocket, two hydrogen bonds can form spontaneously, S86–L28 and T30–W89, with S86 and T30 orienting their hydrophilic hydroxyl groups away from the cavity. Alternatively, the crucial hydrogen bond with W89 can also be established with the neighbouring T31 side chain (Supplementary Fig. 14). Flanking the p2 pocket, interaction of P33 with F94 leads to a reduction in hydrophobic surface area and may contribute favourably to stability of the holo state of e1 in water. Further away from p2, long-range changes involving e1 residues R34 and S36 become apparent in the simultaneous breaking of two interstrand backbone hydrogen bonds (S36–D76 and Y38–F74) and their re-formation with new partners (R34–D76 and S36–F74) in a 'register shift' motion (Fig. 5b), in full accordance with the observations by cryo-EM (Fig. 4c,d).

Two sets of control simulations of S1$^A$ allowed us to infer a specific role of the ligand in the observed S1$^A$ dynamics (Extended Data Fig. 10). In keeping with the inherent flexibility of the e1 loop, all individual e1 interactions can indeed also occur in the absence of the ligand. Without the ligand, however, these interactions remained highly dynamic. Yet, when the ligand encountered the alternative e1 state, either 'naturally' during the simulations or by simulations of a pre-built complex resembling the 'holo' cryo-EM structure, this pattern changed substantially.

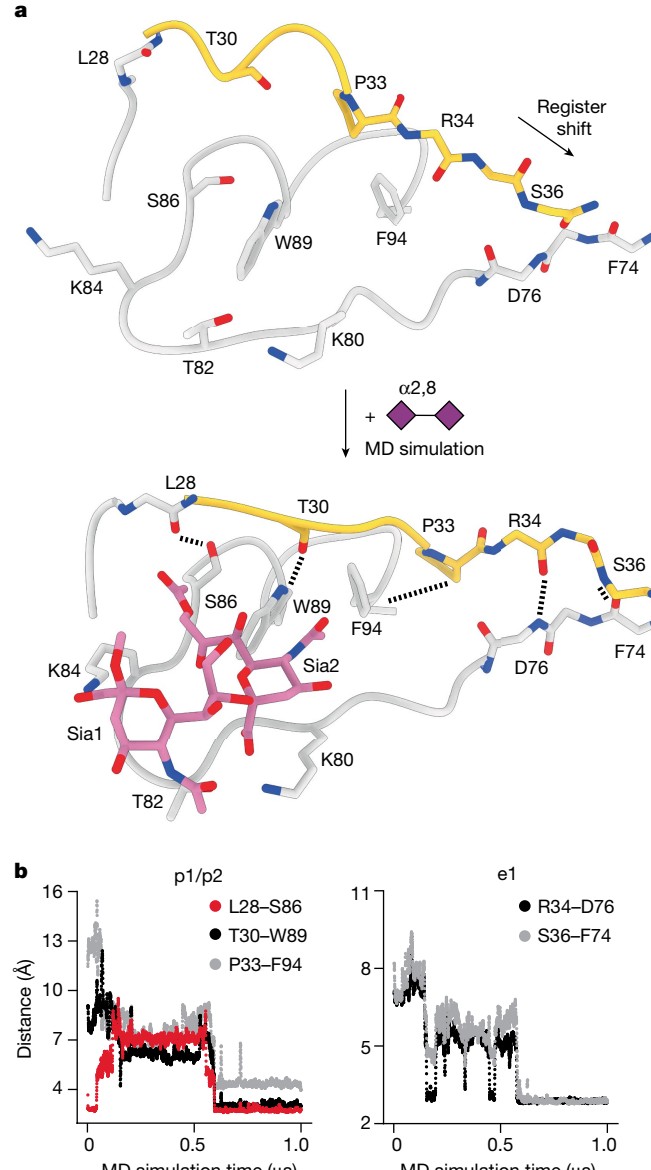

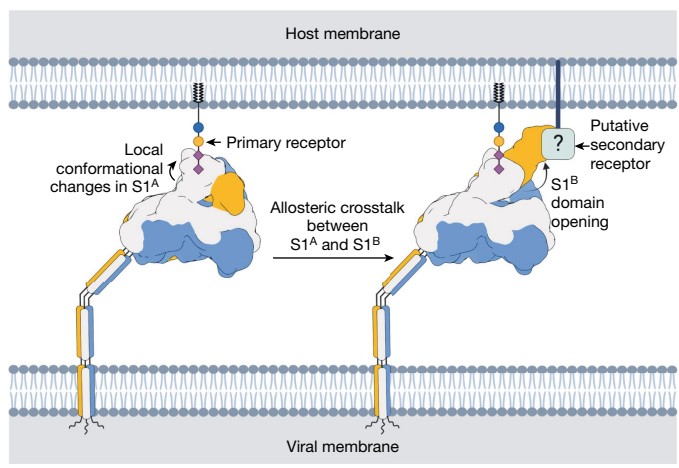

**Fig. 6 | Proposed model for HKU1-A spike host cell engagement.** The HKU1-A spike engages a primary carbohydrate receptor, containing a 9-*O*-acetylated α2,8-linked disialoside, through the S1$^A$ domain. This causes the allosteric opening of the neighbouring S1$^B$ domain, which then binds to a putative secondary receptor. Created with BioRender.com.

**Fig. 5 | MD analysis predicts S1$^A$ conformational transition. a**, An exemplar molecular dynamics simulation trajectory. Docking of the disialoside (purple diamonds) into the apo cryo-EM model (top) converts S1$^A$ into the stable holo state (bottom). **b**, New key hydrogen bonds and hydrophobic contacts form within 500 ns, altering the topologies of the e1 loop (shown in yellow in **a**) and the p1 and p2 pockets. Changes in inter-residue distances (*y* axis) plotted against time (*x* axis). See also Supplementary Video 7.

The hallmark interactions, including the signature register shift in the S1$^{A1}$–S1$^{A2}$ hinge, reproducibly remained stable for several hundred nanoseconds. The collective results of cryo-EM and molecular dynamics analyses indicate that ligand binding stabilizes the shifted topology of the e1 element, apparently locking subdomain S1$^{A1}$ in a state that allows subsequent conformational S1$^B$ changes to occur.

## Discussion

The dynamic sampling of open and closed conformations by sarbecovirus and merbecovirus spike proteins has become emblematic of how CoVs would balance host cell attachment and immune escape. The transition to the open state exposes subdomain S1$^B$ for its binding to proteinaceous cell surface receptors and is also deemed crucial to

allow protein refolding during S-mediated membrane fusion. Remarkably, however, with rare exception the pre-fusion spike proteins from all other CoVs studied so far have all been observed in the closed state exclusively (Supplementary Table 1). Here we shed new light on this apparent contradiction by demonstrating that the spike protein of a HKU1-A strain can in fact transition into an open state, albeit not spontaneously but on a specific cue. Binding of the disialoside-based receptor 9-*O*-Ac-Sia(α2,8)Sia to S1$^A$ triggers a major shift causing the S1$^B$ subdomain to become exposed in a 1-up and eventually fully open, 3-up conformation. The exposure of S1$^{B2}$ would allow for interactions with a putative secondary receptor and thus adds to the notion that such a receptor exists[23,24]. On the basis of the collective data, we propose a model in which binding to a primary sialoglycan-based receptor triggers opening of S1$^B$, which in turn engages a yet unidentified secondary receptor required for entry (Fig. 6).

Four different spike protien structures were identified that together capture a trajectory from a closed apo to a fully open holo conformation. The initial step, S1$^A$ disialoside binding, converts the protein into a conformationally distinct state, still fully closed but primed for S1$^B$ transition, transient yet stable enough to be detected in our analyses. The binding of the disialoside receptor analogue leads to various structural changes within the S1$^{A1}$ subdomain. Most prominently, it stabilizes an alternative topology of the e1 element, only fleetingly attained in the apo structure. Inward e1 displacement walls off one side of the 9-*O*-Ac-Sia-binding site, deepening the p1 pocket and adding to its hydrophobicity. Accommodation of the sialate-9-*O*-acetyl within the p1 pocket may well act as the nucleating event from which other conformational changes follow. These extend to a distal hinge element that connects the S1$^{A1}$ and S1$^{A2}$ subdomains.

Our findings suggest a causal mechanistic relationship between the disialoside-induced conformational changes in e1, S1$^{A1}$ rotation, the remodelling of the S1$^A$–S1$^B$ interface and S1$^B$ expulsion. Yet, we note that the topology of the e1 element in our HKU1-A spike apo structure is atypical and differs from that in the spike protein of HKU1-B and those of betacoronavirus-1 variants OC43, bovine CoV and porcine haemagglutinating encephalomyelitis virus[15,16,27,32] (Supplementary Fig. 15). In the apo structures of these other proteins, the extended e1 element already adopts the topology of that in the HKU1-A closed holo structure. Moreover, in the HKU1-B spike apo structure, subdomains S1$^{A1}$ and S1$^{A2}$ are in similar spatial juxtaposition as in the A-type spike holo conformation. Under the assumption that the other embecovirus

spike proteins also transition into an open conformation, they might do so through a distinct allosteric mechanism. However, given that cryo-EM models are based on averaging, it is quite possible that also in the HKU1-B and betacoronavirus-1 spike proteins the e1 element continuously samples both topologies. If so, the transition of S1[B] into the up position may critically depend on an increase in the lifetime of the shifted state as induced by S1[A] ligand binding. The difference between the A- and B-type spike proteins in their preferred apo topologies of the e1 element may have arisen from immune selection. Indeed, we recently demonstrated that the S1[A] receptor-binding site of OC43, which exhibits the shifted topology, is targeted by potent neutralizing antibodies[22].

The question remains why the transition into S1[B] up conformations was not observed in our previous study of an OC43 S–receptor complex[16]. Possibly, the 9-O-Ac-Sia monosaccharide that was used as a receptor analogue does not suffice to trigger the conformational changes and a more complex glycan may be required. Of note, OC43 spike binds to α2,3- and α2,6-linked 9-O-Ac-sialosides[16], but exhibits a preference for 9-O-Ac-Sia(α2,8)Sia[17]. Evidence that OC43 spike proteins can indeed transition to an open state with S1[B] exposure comes from our recent observation of neutralizing antibodies targeting cryptic S1[B] epitopes. Moreover, virus neutralization by these antibodies selected for resistance mutations in the e1 loop of S1[A] (ref. 22). These results align with our present observations for HKU1, indicating that there is allosteric crosstalk between the S1[A] and S1[B] domains shared among embecoviruses. Hypervariable S1[A] loop elements controlling both S1[B] opening and S2′ proteolytic processing, as described for SARS-CoV-2, might even indicate that this is a universal feature of (beta)coronavirus spike proteins[33,34]. In this view, sarbecoviruses and merbecoviruses spontaneously exposing S1[B] would not be exceptions but part of a mechanistic spectrum, with other CoVs, such as HKU1, relying on specific triggers such as binding to primary receptors by S1[A]. To our knowledge, this is the first description of a CoV spike protein exposing its S1[B] domain on cue. Our observations suggest that CoV attachment may be even more sophisticated than appreciated so far, with possibilities of dual receptor usage and priming of entry to escape immune detection.

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

# Methods

## Expression and purification of trimeric HKU1 spike ectodomains

The sequence of a HKU1-A spike protein (GenBank: ADN03339.1) coding for the ectodomain (residues 12–1266) was cloned into the pCG2 expression vector with an exogenous CD5 signal peptide. At the 3′ end, the coding sequence was ligated in frame with a GCN4 trimerization motif (IKRMKQIEDKIEEIESKQKKIENEIARIKKIK)[35,36], a thrombin cleavage site (LVPRGSLE), an 8-residue long Strep-Tag (WSHPQFEK) and a stop codon. The furin cleavage site at the S1/S2 junction was mutated from RRKRR to GGSGS to avert cleavage of the spike protein (Supplementary Fig. 17). The resulting construct was used for transient expression in HEK293T cells and purified as previously described[37]. In brief, after incubation of the cells for 5 days, spike glycoprotein was purified from cleared cell culture supernatants by affinity chromatography using StrepTactin beads (IBA) and eluted in 20 mM Tris-HCl, pH 8.0, 150 mM NaCl, 1 mM EDTA, 2.5 mM D-biotin. The W89A mutant protein was produced as described previously[17].

## Sample preparation for cryo-EM

For the apo complex, 3 µl of 4.3 µM HKU1 spike trimer was applied to QuantiFoil R1.2/1.3 grids that had been glow-discharged for 30 s on a GloQube (Quorum) at 20 mW power. The sample was applied at 4 °C and 95% relative humidity inside a Vitrobot Mark IV (Thermo Scientific). The grids were then blotted for 7 s with +2 blot force and plunge-frozen in liquid ethane. For the holo complex and W89A negative control, 7 µl of 4.3 µM wild-type or mutant HKU1 spike trimer was combined with 3 µl of 1 mM sugar, resulting in a final spike protein concentration of 3 µM and sugar concentration of 300 µM. The samples were then incubated at room temperature for about 10 min before vitrification, which was carried out as described for the apo sample.

## Cryo-EM data acquisition

The apo and holo HKU1 spike samples were imaged on a Thermo Scientific Krios G4 Cryo-TEM equipped with a K3 direct electron detector and a BioContinuum energy filter (Gatan) using EPU 2 acquisition software. The stage was pre-tilted to 30° to improve the orientation distribution of the particles. A total of 4,207 videos for apo spike and 4,065 videos for the holo spike were collected at a super-resolution pixel size of 0.415 Å per pixel, with 40 fractions per video and a total dose of 46 electrons per Å$^2$. Defocus targets cycled from −1.5 to −2.5 µm.

The W89A mutant HKU1 spike incubated with disialoside was imaged on a Thermo Scientific Glacios cryo-TEM instrument equipped with a Falcon 4 direct electron detector using EPU 2 acquisition software. The stage was pre-tilted to 30° to improve the orientation distribution of the particles. A total of 896 videos were collected at 0.92 Å per pixel with 40 fractions per video and a total dose of 42 electrons per Å$^2$. Defocus targets cycled from −1.5 to −2.5 µm. A summary of all data collection parameters is shown in Supplementary Table 2.

## Single-particle image processing

For the apo complex, patch motion correction, using an output F-crop factor of 0.5, and patch CTF estimation were carried out in cryoSPARC live[38]. Micrographs with a CTF estimated resolution of worse than 10 Å were discarded, leaving 4,202 images for further processing. The blob picker tool was then used to select 9,144,772 particles that were then extracted in a 100-pixel box (Fourier binned 4 × 4) and then exported to cryoSPARC for further processing. A single round of two-dimensional (2D) classification was carried out, after which 183,886 particles were retained. Ab initio reconstruction generated one well-defined reconstruction of the closed HKU1 spike protein. Particles belonging to this class were then re-extracted in a 300-pixel box. During extraction, particles were Fourier binned by a non-integer value, resulting in a final pixel size of 1.1067 Å. Subsequently, non-uniform refinement was carried out on the extracted particles with $C_3$ symmetry imposed[39],

yielding a reconstruction with a global resolution of 3.3 Å. Subsequently, each particle from the $C_3$-symmetry-imposed reconstruction was assigned three orientations corresponding to its symmetry-related views using the symmetry expansion job. A soft mask encompassing one S1$^A$ domain was made in UCSF Chimera[40], and used for local refinement of the expanded particles, yielding a map with a global resolution of 3.8 Å.

For the holo complex, patch motion correction, using an output F-crop factor of 0.5, and patch CTF estimation were carried out in cryoSPARC live[38]. Micrographs with a CTF estimated resolution of worse than 10 Å were discarded, leaving 4,045 images for further processing. The blob picker tool was then used to select 956,697 particles that were then extracted in a 100-pixel box (Fourier binned 4 × 4) and then exported to cryoSPARC for further processing. Four parallel rounds of 2D classification were carried out, using an initial classification uncertainty value of 1, 2, 4 or 6. Subsequently, the well-defined spike classes were selected from each 2D run and combined. Duplicate particles were then removed, after which 169,728 particles were retained. Ab initio reconstruction generated two classes corresponding to the closed and 3-up spike trimer. Particles from these two classes were used as the input for a second round of ab initio reconstruction that produced two classes corresponding to the 3-up and 1-up spike trimer, although the latter seem to be a convolution of 1-up and closed particles. These two volumes were then used as initial models for a round of heterogeneous refinement. To avoid missing spike particles that may have been removed during initial stringent selection of 2D classes, heterogeneous refinement was carried out on a larger particle stack of 895,888 particles, from which only carbon classes had been removed from the initial stack. Heterogeneous refinement produced two well-defined reconstructions of the 3-up and 1-up conformations. Particles corresponding to the 3-up class were subjected to a single round of 2D classification and the clearly defined spike protein classes were selected. These were then re-extracted in a 300-pixel box. During extraction, particles were Fourier binned by a non-integer value, resulting in a final pixel size of 1.1067 Å. Subsequently, non-uniform refinement was carried out on the extracted particles with $C_3$ symmetry imposed[39], yielding a reconstruction with a global resolution of 3.7 Å. As a result of the apparent heterogeneity in the 1-up sample, an additional round of heterogeneous refinement was carried out on the 895,888-particle stack, using higher-quality initial models, namely the fully refined 3-up map and the 1-up map obtained from the second round of ab initio reconstruction. Heterogeneous refinement produced well-defined reconstructions of the 3-up and 1-up conformations. Particles corresponding to both classes were individually subjected to a single round of 2D classification and the clearly defined spike classes were selected. These were then individually re-extracted in a 300-pixel box. During extraction, particles were Fourier binned by a non-integer value, resulting in a final pixel size of 1.1067 Å. Subsequently, non-uniform refinement was carried out on the extracted particles with $C_3$ or $C_1$ symmetry imposed, yielding reconstructions with global resolutions of 3.56 and 4.13 Å for the 3-up and 1-up conformations, respectively. After global refinement, a soft mask encompassing one S1$^A$ domain of the 3-up sample was made in UCSF Chimera. Local refinement was then carried out on the 3-up particles, yielding a map with a global resolution of 4.19 Å. The particles belonging to the 1-up reconstruction were subjected to another round of heterogeneous refinement, which produced two clear reconstructions of the closed and 1-up spike protein. Non-uniform refinement was carried out on both sets of particles with $C_3$ or $C_1$ symmetry imposed, yielding reconstructions with global resolutions of 3.68 and 4.68 Å for the closed and 1-up conformations, respectively. For the closed spike protein, each particle from the $C_3$-symmetry-imposed reconstruction was assigned three orientations corresponding to its symmetry-related views using the symmetry expansion job. A soft mask encompassing one S1$^A$ domain was made in UCSF Chimera[40], and the symmetry-expanded particles were subjected to masked 3D

variability analysis[41]. Local refinement was then carried out on the particles belonging to the best resolved cluster, yielding a map with a global resolution of 4.13 Å.

For the W89A mutant HKU1 spike incubated with disialoside, patch motion correction was carried out in MotionCor2 (ref. 42), implemented through Relion version 3.1.1 (ref. 43). The motion-corrected micrographs were then imported into cryoSPARC for patch CTF estimation and further processing steps[38]. The blob picker tool was used to select 215,843 particles that were then extracted in a 100-pixel box (Fourier binned 4 × 4). A single round of 2D classification was carried out, after which 38,838 particles were retained. Ab initio reconstruction generated one well-defined reconstruction of the closed HKU1 spike protein. Particles belonging to this class were then re-extracted in a 300-pixel box. During extraction, particles were Fourier binned by a non-integer value, resulting in a final pixel size of 1.2267 Å. Subsequently, non-uniform refinement was then carried out on the extracted particles with $C_3$ symmetry imposed[39], yielding a reconstruction with a global resolution of 5.1 Å. Subsequently, each particle from the $C_3$-symmetry-imposed reconstruction was assigned three orientations corresponding to its symmetry-related views using the symmetry expansion job. A soft mask encompassing one S1$^A$ domain was then made in UCSF Chimera[40], and used for local refinement of the expanded particles, yielding a map with a global resolution of 5.4 Å.

The 'gold standard' Fourier shell correlation (FSC) criterion (FSC = 0.143) was used for calculating all resolution estimates, and 3D-FSC plots were generated in cryoSPARC[44]. To facilitate model building, globally refined maps were sharpened using DeepEMhancer (version 0.13)[45], as implemented in COSMIC2[46], or filtered by local resolution in cryoSPARC.

## Modelling

Initially, a homology model for HKU1-A spike protein was generated by Phyre 2 (ref. 47) with the embecovirus OC43 spike structure (Protein Data Bank (PDB) 6NZK)[24] as template. The HKU1-A spike homology model was rigid body fitted into the apo-state cryo-EM map using the UCSF Chimera[40] tool Fit in map. The crystal structure of HKU1-A S1$^B$ (PDB 5KWB; ref. 24) was used to replace the equivalent S1$^B$ domain in the homology model owing to clearly wrong homology modelling. Models were refined by carrying out iterative cycles of manual model building using Coot[48] and real-space refinement using Phenix[49]. The Coot carbohydrate module[50] was used for building N-linked glycans, which were manually inspected and corrected. The apo state was modelled first, owing to its highest resolution. Subsequently, the closed holo, the holo 3-up and the holo 1-up were modelled in that order, using previous models as a starting point. For the initial holo (closed) S1$^A$ model, Namdinator[51] was used for flexible fitting in a locally refined and unsharpened map for the closed holo S1$^A$. Model validation was carried out using Molprobity and Privateer[52–54].

Elbow[55] was used to generate ligand restraints for the 9-$O$-acetylated terminal sialic acid based on the 'MJJ' ligand in the OC43 spike cryo-EM structure (PDB 6NZK)[16], after which atom names were manually modified to be consistent with the earlier standard MJJ model and general sialic acid atom numbering, and the O2-attached methyl linker atoms of the original MJJ ligand were trimmed. As there is no standard MJJ–SIA α2,8 linkage defined in software packages used at present, we used molecular dynamics-based restraints (see below) to model this glycosidic linkage of the disialoside. The following restraints were used for the glycosidic linkage between the terminal 9-$O$-acetylated sialic acid (ligand code MJJ) and the penultimate sialic acid (ligand code SIA) based on the most common solution conformer: bond distance C2–O8 of 1.38 Å ($\sigma$ of 0.01 Å); bond angles of 109.5° for O8–C2–O6 and for O8–C2–C3, and 114.5° for O8–C2–C1 (all $\sigma$ of 2.0°); dihedral angles of 295.0° for C1–C2–O8–C8 and of 122° for C2–O8–C8–C7 (both $\sigma$ of 5.0°).

## MD simulations

Starting structures of the molecular systems were built on the basis of the cryo-EM structures of HKU1 (this work) using the graphical interface of YASARA[56]. The N-glycans were attached to the protein on the basis of data from quantitative site-specific N-linked analysis of HKU1 spike protein[31]. Models of the complexes with α-Neu5,9Ac-(2-8)-α-Neu5Ac-OMe were built on the basis of the holo and apo versions of S1$^A$ (residues 14–299). The ligand was positioned manually into the binding site guided by interactions found in PDB entry 6NZK (hCoV-OC43). The HKU1 N1 sequence was taken from GenBank entry NC_006577.2.

Each system was positioned in a periodic rectangular cuboid simulation box (10 Å buffer around the solute) and the AMBER14 force field was selected, which uses ff14SB (ref. 57) and GLYCAM06j (ref. 58) parameters (including mixed 1–4 scaling). YASARA offers several automated workflows (termed experiments) for system setup. The 'neutralization experiment' was used to adjust the protonation states of the amino acids (pH 7.4)[59] and to solvate the system in 0.9% NaCl solution (0.15 M). The 'minimization experiment' (short steepest descent minimization followed by simulated annealing minimization until convergence is reached) was used to remove conformational stress in the system. Simulations were carried out at 310 K using periodic boundary conditions and the particle mesh Ewald algorithm[60] to treat long-range electrostatic interactions. Temperature was rescaled using a tuned Berendsen thermostat[61]. The box size was rescaled dynamically to maintain a water density of 0.996 g ml$^{-1}$ ('densostat' method for pressure coupling)[62]. Position restraints were active during the equilibration phase for at least 3 ns (at the beginning, on all protein heavy atoms, and then only on backbone atoms). To prevent dissociation of the ligand, distance restraints were applied to maintain the critical H bonds for binding (between atoms K80:O and SIA_2:N5; K80:NZ and SIA_2:O1A; T82:OG1 and SIA_2:O1B). Production simulations were carried out using YASARA with GPU acceleration in 'fast mode' (mixed multiple time-step algorithm reaching 5 fs)[62] on 'standard computing boxes' equipped, for example, with one 12-core i9 CPU and NVIDIA GeForce GTX 1080 Ti. Harmonic position restraints (stretching force constant = 1 N m$^{-1}$) were applied to protein backbone atoms of residues 48–65 and 264–299 of the S1$^A$ system to prevent system rotation in the cuboid box and to deal with the 'artificially loose end' at residue 299. The average root mean square deviation of the protein Cα atoms was monitored to check the overall stability of the simulation.

To visualize the glycan coverage of the closed spike protein, the fully glycosylated ectodomain system (590,814 atoms) was simulated with position restraints on backbone atoms of residues 1080–1110 for 250 ns with a performance of about 4 ns per day. Molecular systems based on S1$^A$ alone were smaller (approximately 32,500–56,200 atoms, depending on the size of the N-glycans attached) and were sampled for an accumulated timescale of approximately 20 μs for the Caen1 sequence (apo + disialoside ligand, 5 μs, 6 simulations; holo, 15 μs, 27 simulations) and 52 μs for the N1 sequence (apo, 12 μs, 13 simulations; apo + disialoside ligand, 23 μs, 34 simulations; holo, 17 μs, 22 simulations) with individual simulations reaching up to 1.6 μs. The performance was about 100–200 ns per day. Distances shown in Fig. 5b were calculated from an example trajectory (Extended Data Fig. 10a) between the following atoms: L28:O and S86:OG; P33:CG and F94:CA; T30:O and W89:NE1; S36:N and F74:O; R34:O and -D76:N. Additionally, the solvated disialoside ligand was simulated without the protein using YASARA (general molecular dynamics parameters used as described above) in a cubic box with side length of 37 Å for 10 μs at 310 K using GLYCAM06j parameters. These simulation data were used to identify low-energy conformers of the disialoside ligand, which were used to support the modelling of the reducing-end Neu5Ac residue into the local low-resolution cryo-EM density.

Conformational Analysis Tools (http://www.md-simulations.de/CAT/) was used for analysis of trajectory data, general data processing

and generation of scientific plots. VMD[63] was used to generate molecular graphics.

## Analysis and visualization

Spike interface areas were calculated using PDBePISA[64]. Surface colouring of HKU1-A spike protein according to sequence conservation was carried out using Consurf[65] and visualized in UCSF ChimeraX[66]. The UCSF Chimera MatchMaker tool was used to obtain root mean square deviation values, using default settings. Domain rotations were calculated with CCP4 (ref. 67) Superpose[68]. Figures were generated using UCSF ChimeraX[66] and BioRender.com. Structural biology applications used in this project were compiled and configured by SBGrid[69].

## Reporting summary

Further information on research design is available in the Nature Portfolio Reporting Summary linked to this article.

## Data availability

The atomic models of the apo, holo, 1-up and 3-up HCoV-HKU1 spike have been deposited to the Protein Data Bank (PDB) under the accession codes 8OHN, 8OPM, 8OPN and 8OPO. The globally and locally refined cryo-EM maps have been deposited to the Electron Microscopy Data Bank (EMDB) under accession codes EMD-16882, EMD-17076, EMD-17077, EMD-17078, EMD-17079, EMD-17080, EMD-17081, EMD-17082 and EMD-17083. Data files pertaining to molecular dynamics simulation results shown in Fig. 5b and Extended Data Figs. 7–10 are available at https://doi.org/10.5281/zenodo.7867090.

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

**Acknowledgements** We thank R. Dijkman for providing the HKU1 Caen1 sequence and J. de Groot-Mijnes for critically reading the manuscript. We are grateful for computer time provided by BIOGNOS AB, Göteborg. This work was supported by the China Scholarship Council 2014-03250042 (Y.L.). This work made use of the Dutch national e-infrastructure with the support of the SURF Cooperative using grant no. EINF-2453, awarded to D.L.H. R.C. acknowledges funding by the Deutsche Forschungsgemeinschaft (494746248). R.J.G.H. is financially supported by a Dutch research council NWO-XS grant (OCENW.XS22.3.110). G.-J.B. is supported by an ERC advanced grant (SWEETPROMISE, 101020769); M.F.P. and D.L.H. are supported by NWO Veni grants (VI.Veni.202.271 and VI.Veni.212.102, respectively). J.S. is financially supported by the Dutch Research Council NWO Gravitation 2013 BOO, Institute for Chemical Immunology (024.002.009).

**Author contributions** Y.L., R.J.d.G. and D.L.H. conceived the project; Y.L., M.F., R.J.d.G. and D.L.H. designed the experiments; Y.L. designed and cloned the protein constructs and carried out protein expression and purification; I.D., Z.L., F.J.M.v.K., J.S., B.-J.B., G.-J.B. and M.F. provided access to equipment and reagents; I.D. carried out cryo-EM sample preparation and data collection; D.L.H. processed the cryo-EM data; M.F.P. and D.L.H. built and refined the atomic models. M.F. carried our molecular dynamics simulations; M.F.P., R.C., M.F., R.J.d.G. and D.L.H. analysed and visualized the data; M.F.P., R.C., M.F. and D.L.H. curated the data. R.J.d.G. and D.L.H. supervised the project. M.F.P., R.C., R.J.G.H., R.J.d.G. and D.L.H. carried out project administration. M.F.P., Y.L., R.J.G.H., R.J.d.G. and D.L.H. obtained funding. M.F.P., R.C., R.J.G.H., R.J.d.G. and D.L.H. wrote the first draft of the manuscript. All authors contributed to reviewing and editing subsequent versions.

**Competing interests** I.D. is an employee of Thermo Fisher Scientific and M.F. is an employee of Biognos AB. The remaining authors declare no competing interests.

## Additional information

**Correspondence and requests for materials** should be addressed to Raoul J. de Groot or Daniel L. Hurdiss.

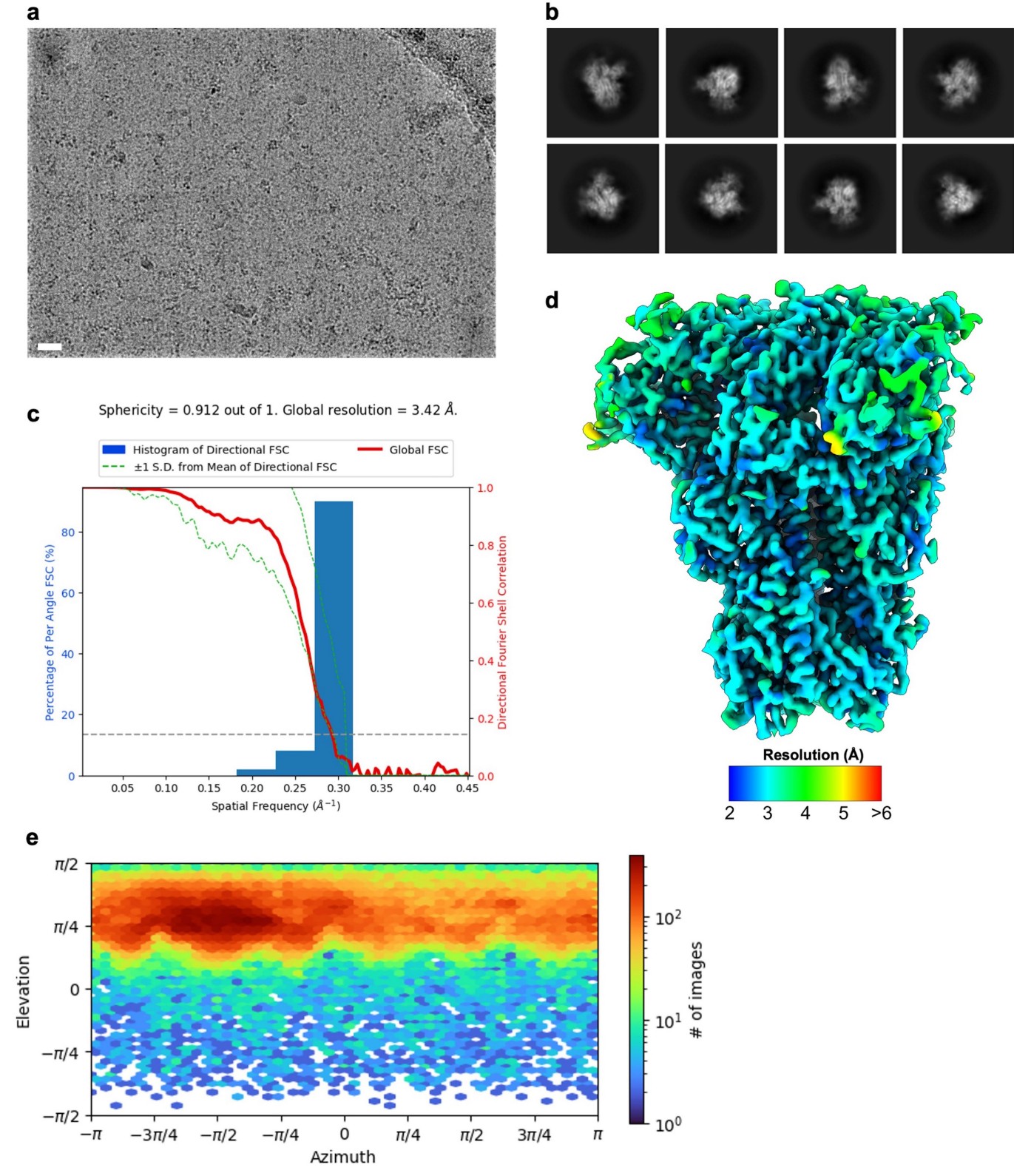

**a**

**b**

**c**

Sphericity = 0.912 out of 1. Global resolution = 3.42 Å.

**d**

Resolution (Å)

2   3   4   5   >6

**e**

**Extended Data Fig. 1 | Cryo-EM data processing of the *apo* HKU1-A spike ectodomain. a**, Representative motion-corrected micrograph out of ~4,200 similar micrographs. Scale bar = 50 nm. **b**, Representative reference-free 2D class averages generated in cryoSPARC. **c**, 3DFSC plot for the 3.4 Å resolution globally refined reconstruction. **d**, DeepEMhancer filtered EM density map for the *apo* HKU1-A spike ectodomains coloured according to local resolution which was calculated in cryoSPARC. **e**, Angular distribution plot calculated in cryoSPARC for particle projections in the globally refined map.

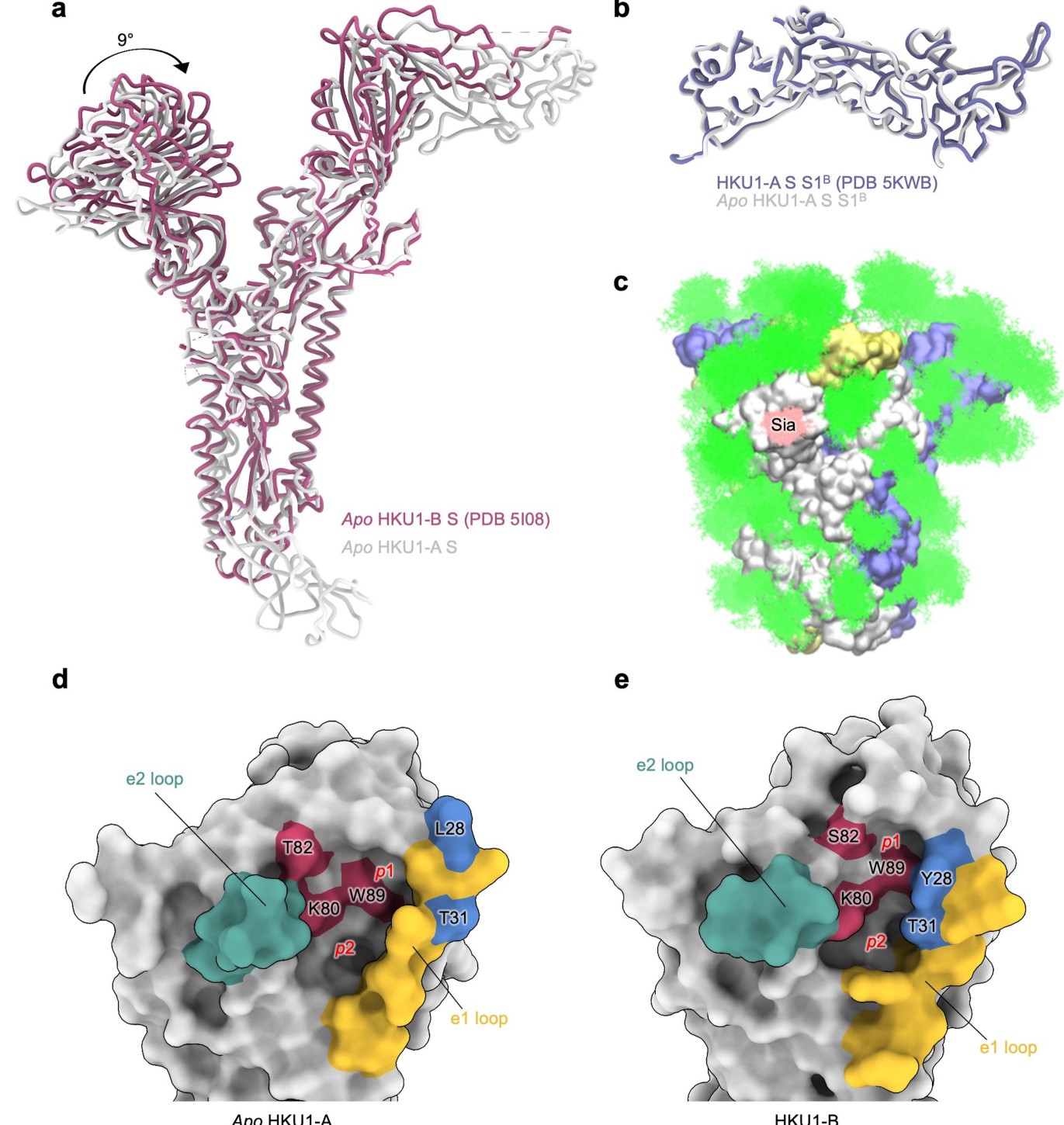

**Extended Data Fig. 2 | Comparison of our *apo* HKU1-A spike structure with previously published structures and visualisation of its glycan shield.**
**a**, Comparison of our *apo* HKU1-A spike (S) structure (in grey) with the previously published structure of full-length HKU1-B spike (dark pink)[27]. **b**, Comparison of our HKU1-A S1^B domain structure with the previously published HKU1-A S1^B domain crystal structure (purple)[24]. **c**, Molecular dynamics (MD)-derived glycan coverage map of the HKU1 spike ectodomain (250 ns, 310 K). Full *N*-glycans (as shown for chain A, see Supplementary Fig. 16a) were attached based on previously published data[31] where available. The spike protomers are coloured grey, blue and yellow and the *N*-linked glycans and bound disialoside (Sia) are coloured green and pink, respectively. To highlight the dynamics of the *N*-glycans, 250 snapshots extracted at time intervals of 1 ns are shown overlayed. **d**, Surface representation of the *apo* HKU1-A sialic acid binding site. Residues critical for sialic acid binding are coloured ruby and selected e1 loop residues are coloured blue. The location of the *p*1 and *p*2 pockets are indicated. **e**, Surface representation of the HKU1-B sialic acid binding site (PDB ID: 5I08)[27], same colouring as panel **d**.

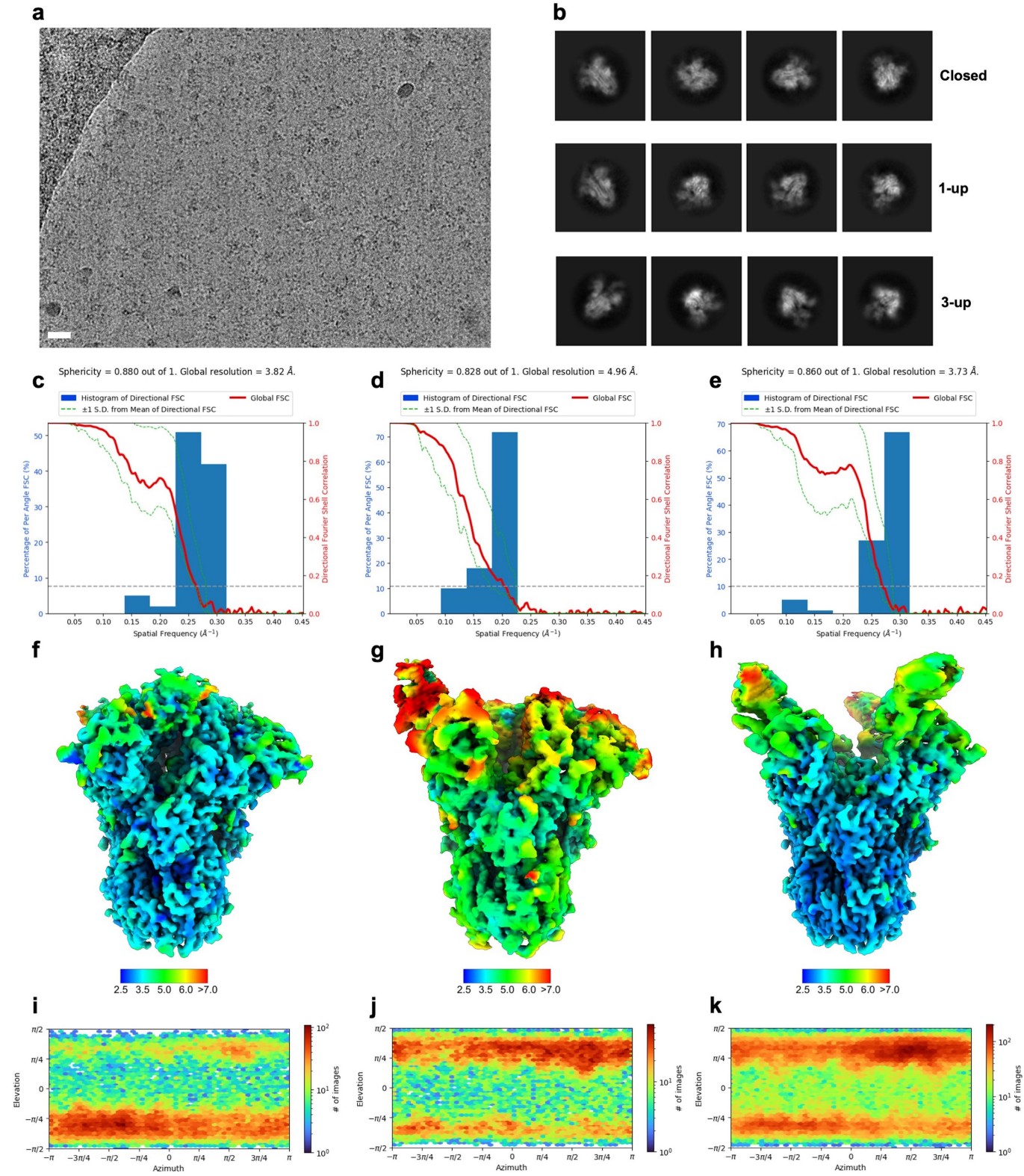

**Extended Data Fig. 3 | Cryo-EM data processing of the *holo* HKU1-A spike ectodomain. a**, Representative motion-corrected micrograph out of ~4,000 similar micrographs. Scale bar = 50 nm. **b**, Representative reference-free 2D class averages of the closed, 1-up and 3-up reconstructions generated in cryoSPARC. **c**, 3DFSC plot for the closed, **d**, 1-up and **e**, 3-up globally refined reconstructions. **f**, DeepEMhancer filtered EM density map for the closed, **g**, 1-up and **h**, 3-up *holo* HKU1-A spike ectodomains coloured according to local resolution which was calculated in cryoSPARC. **i**, Angular distribution plot calculated in cryoSPARC for particle projections in the closed, **j**, 1-up and **k**, 3-up globally refined maps.

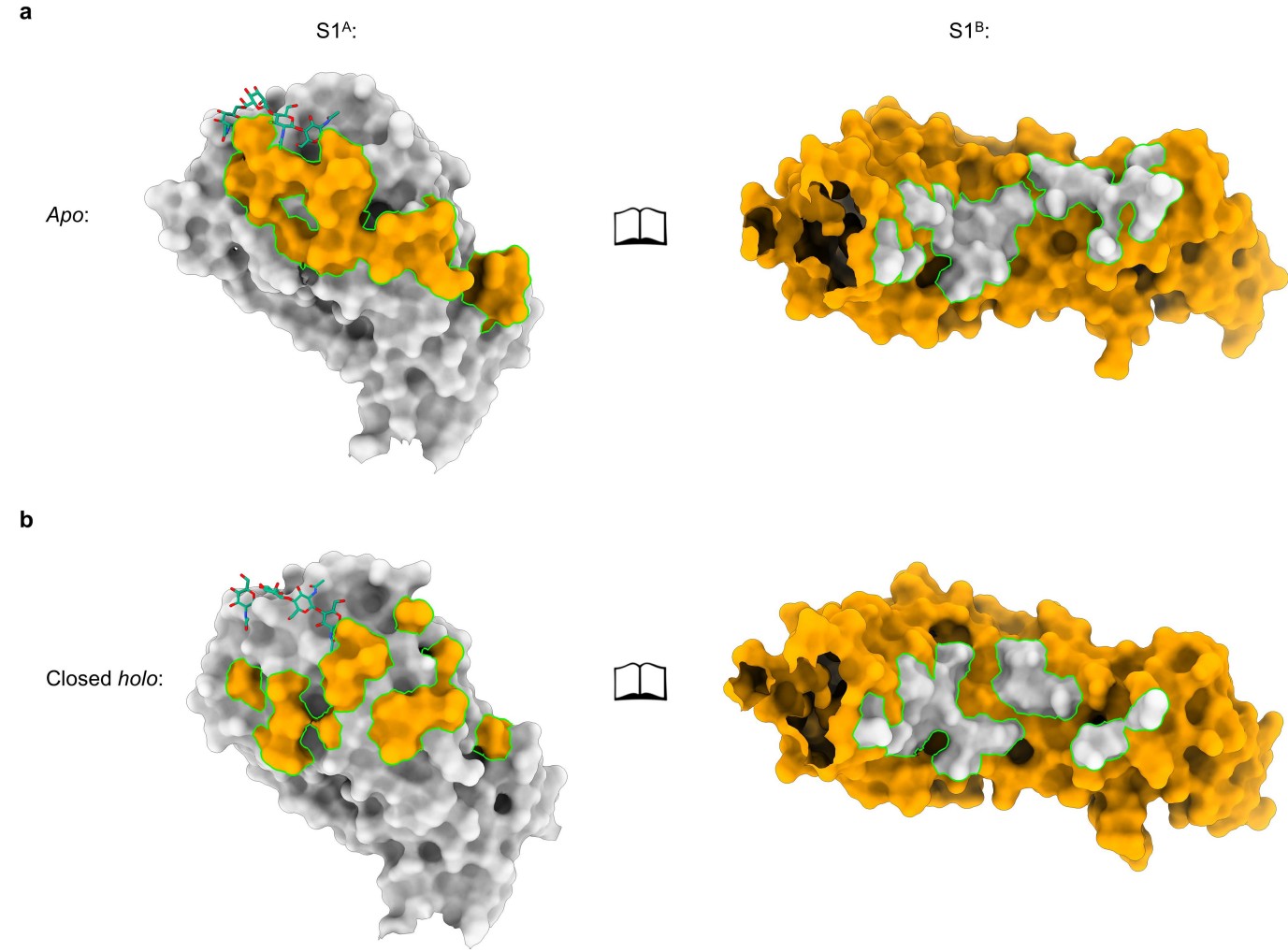

**Extended Data Fig. 4 | Comparison of S1^A-S1^B interface between *apo* and closed *holo* shows a smaller interaction footprint for the latter. a**, Open book representation of the *apo* S1^A-S1^B interface. Interacting surfaces are visualised in the colour of the subunit it interacts with. *N*-linked glycans on S1^A near the interface are indicated as green sticks. **b**, *Idem* for the closed *holo* S1^A-S1^B interface.

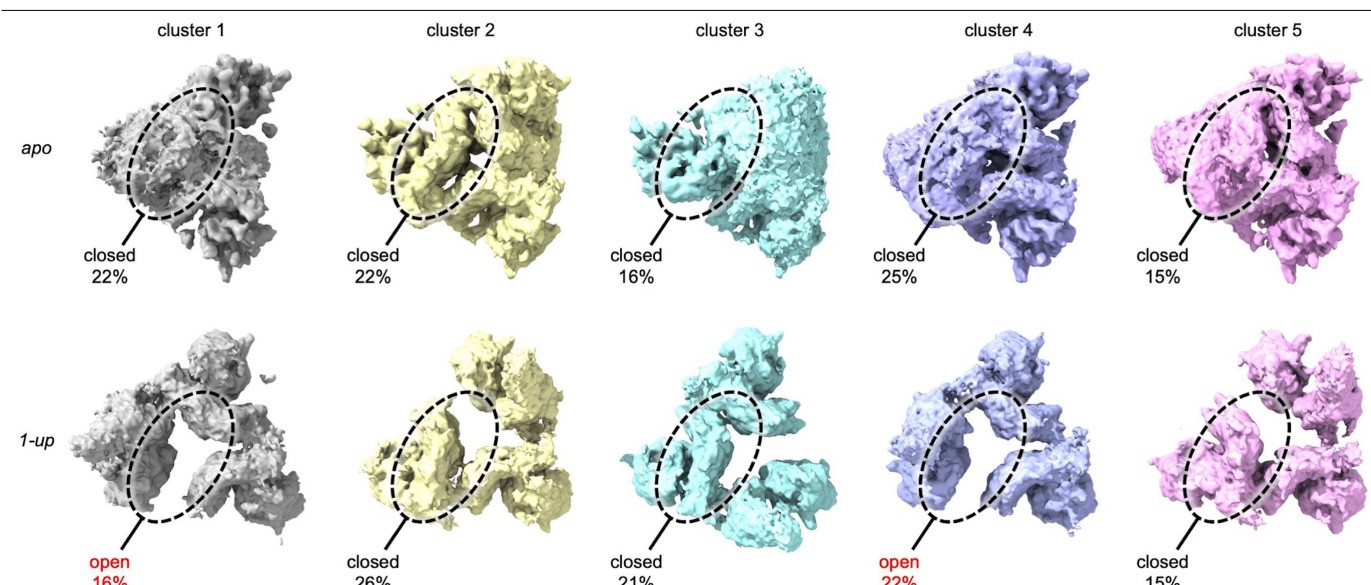

**Extended Data Fig. 5 | 3D variability analysis of the *apo* and 1-up HKU1-A data sets.** 3D variability analysis of the symmetry expanded *apo* HKU1-A particles indicated that there are no detectable open S1<sup>B</sup> domains present in the data. In contrast, this method could discriminate between open and closed S1<sup>B</sup> domains in the *holo* 1-up data set, used as control to show the validity of this approach. The region which was masked during the analysis is circled.

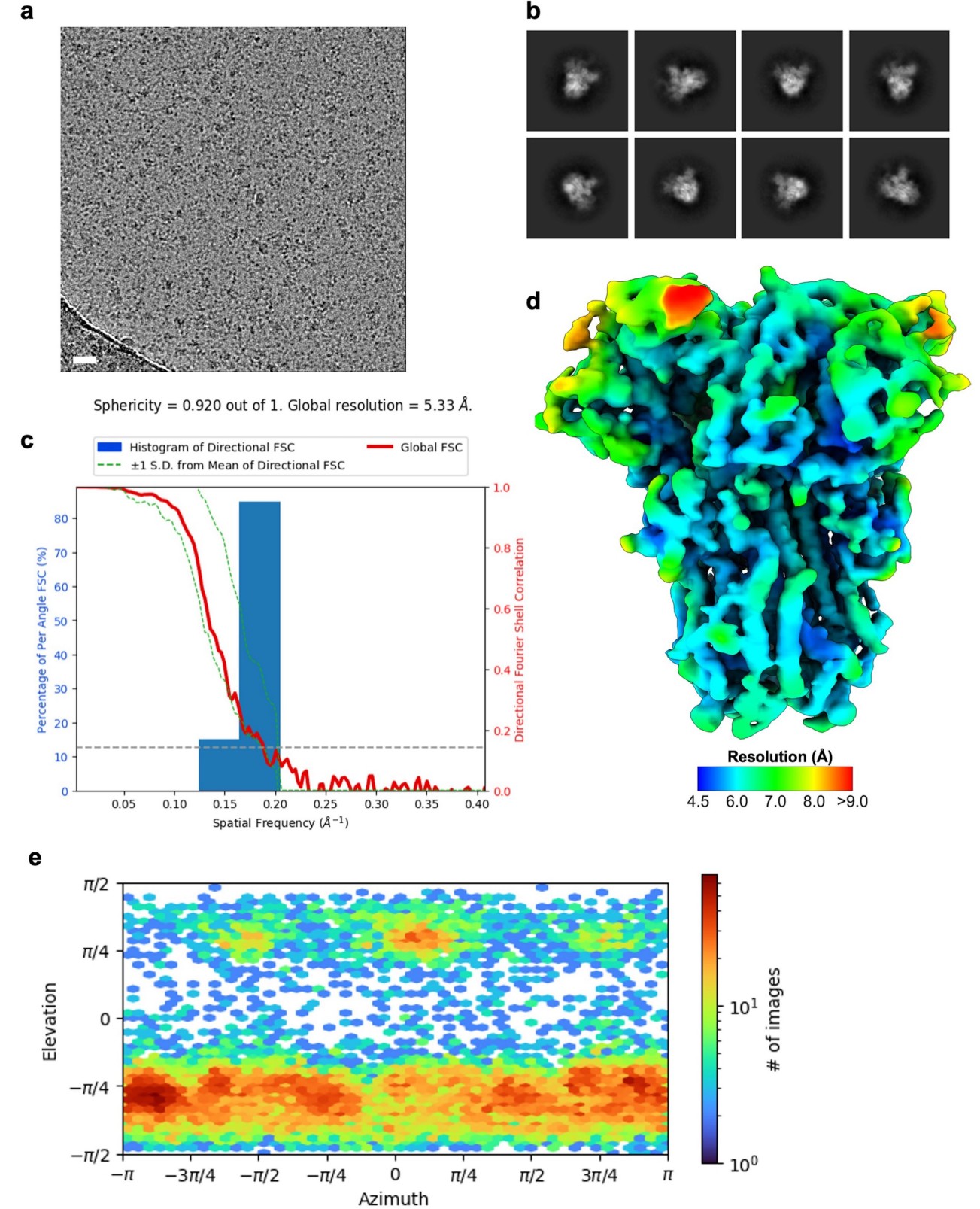

Sphericity = 0.920 out of 1. Global resolution = 5.33 Å.

Resolution (Å)
4.5  6.0  7.0  8.0  >9.0

**Extended Data Fig. 6 | Cryo-EM data processing of the W89A HKU1-A spike ectodomain incubated with disialoside. a**, Representative motion-corrected micrograph out of ~900 similar micrographs. Scale bar = 50 nm. **b**, Representative reference-free 2D class averages generated in cryoSPARC. **c**, 3DFSC plot for the 5.3 Å resolution globally refined reconstruction.

**d**, DeepEMhancer filtered EM density map for the apo HKU1-A spike ectodomains coloured according to local resolution which was calculated in cryoSPARC. **e**, Angular distribution plot calculated in cryoSPARC for particle projections in the globally refined map.

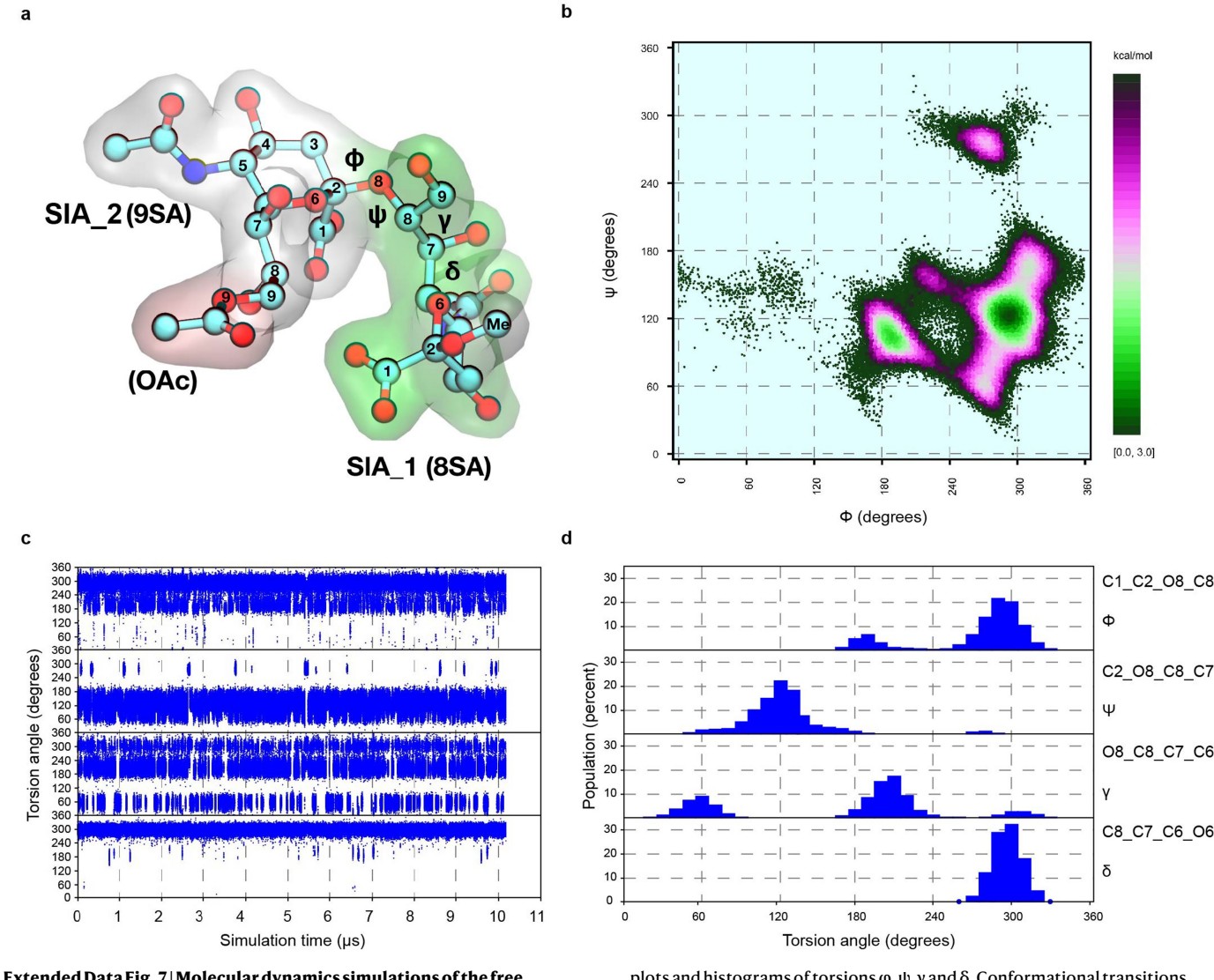

**Extended Data Fig. 7 | Molecular dynamics simulations of the free disialoside.** 10 μs MD-based conformational analysis of Neu5,9Ac$_2$-α2,8-Neu5Ac-α*O*Me in explicit solvent. **a**, Example 3D structure with annotations of residue labels used (GLYCAM residue type labels are shown in brackets), atom numbering scheme and torsions (φ = C1-C2-O8-C8, ψ = C2-O8-C8-C7, γ = O8-C8-C7-C6, δ = C8-C7-C6-O6). **b**, Free energy φ/ψ map. (**c, d**) Trajectory plots and histograms of torsions φ, ψ, γ and δ. Conformational transitions between the population maxima (local energy minima) are fast for φ and γ. Only few transitions occurred for ψ and δ on a 10 μs timescale. Torsion δ has practically only one orientation (about −60°). Data were analysed using Conformational Analysis Tools.

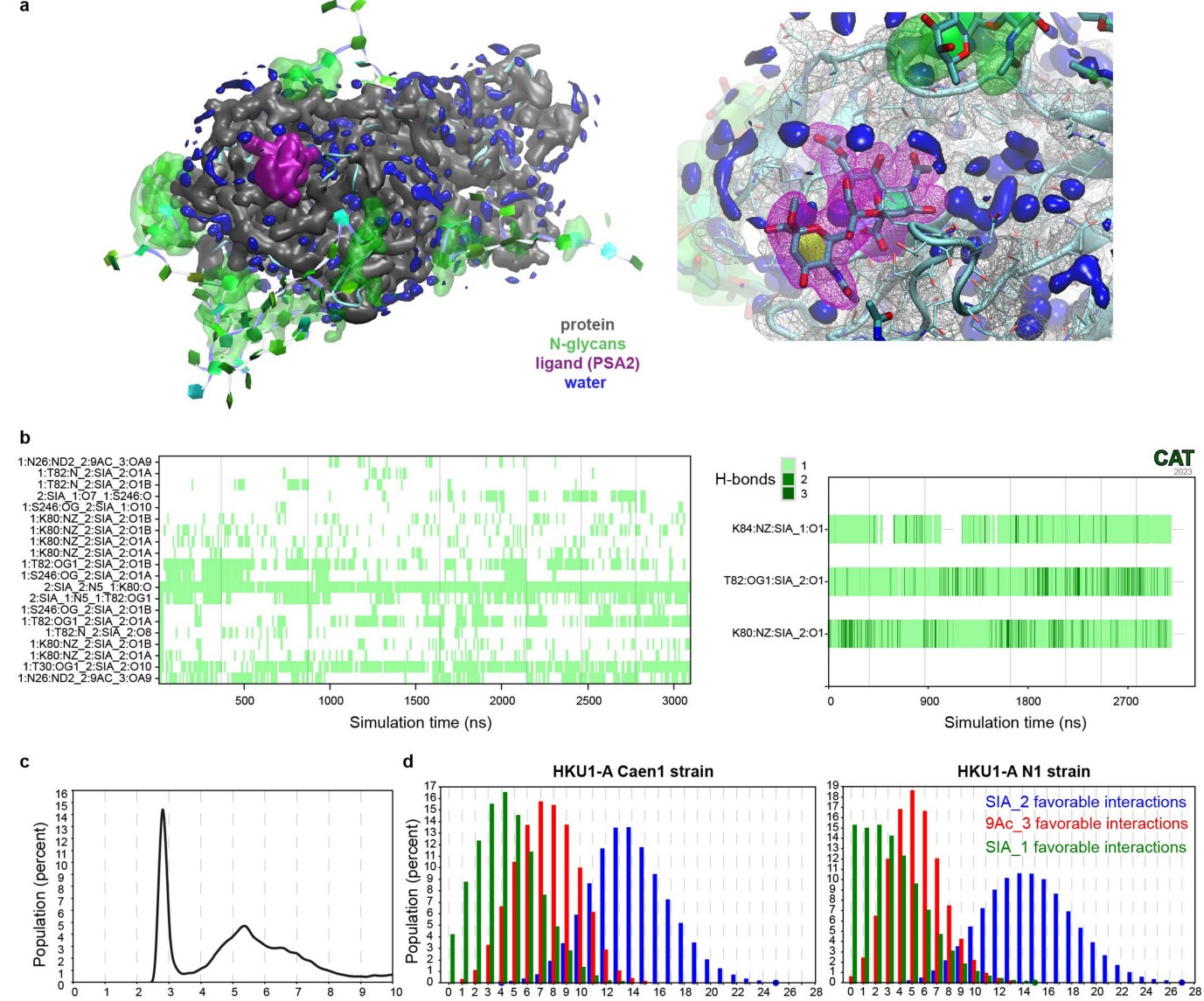

**Extended Data Fig. 8 | MD-derived interactions of the HKU1 spike - α2,8 disialoside complex. a**, MD-derived pseudo-electron density of the disialoside ligand (PSA2, purple) in the binding pocket of the S1^A domain (grey, see supplementary video 6 for a 3D view). Data were derived from 3 µs MD simulations of S1^A (residues 14-299) based on the *holo* cryo-EM model (N-glycans, green, see Supplementary Fig. 16b). **b**, left panel: trajectory plots of the most populated, individual hydrogen bonds between the ligand (molecule 2) and S1^A (molecule 1). Individual simulations are separated by vertical lines. Labels are formatted as follows: donor (D), acceptor (A), *molD:resD:atomD_ molA:resA:atomA*, (see Extended Data Fig. 7a). Donor H atoms were omitted from the labels. A geometric H-bond criterion, defined as distance (D-A) ≤ 3.2 Å and angle (D-H-A) ≥ 120°, was used. Right panel: complex H-bond interactions

of the carboxyl groups of SIA1 and SIA2 involve two equivalent acceptor atoms (O1A and O1B) and potentially multiple, equivalent donor H-atoms (*e.g.* three in Lys:NZ). Trajectory plots show complex H-bond interactions where equivalent H-bonds at a given time were combined and their number is indicated as shades of green. **c**, Histogram of the distance K84(NZ)-SIA1(O1) showing a high probability for a salt bridge between the amino group of K84 and the carboxylate group of SIA1. **d**, Histograms of favourable, stabilising contacts (as defined in Supplementary Table 5) between the individual moieties of the disialoside ligand and the S1^A domains of HKU1 strains Caen1 or N1. A notable decrease in stabilising contacts with the reducing end SIA1 can be seen in HKU1 N1, potentially due to the absence of K84 (see also the hydrogen bond analyses in Supplementary Tables 3–4).

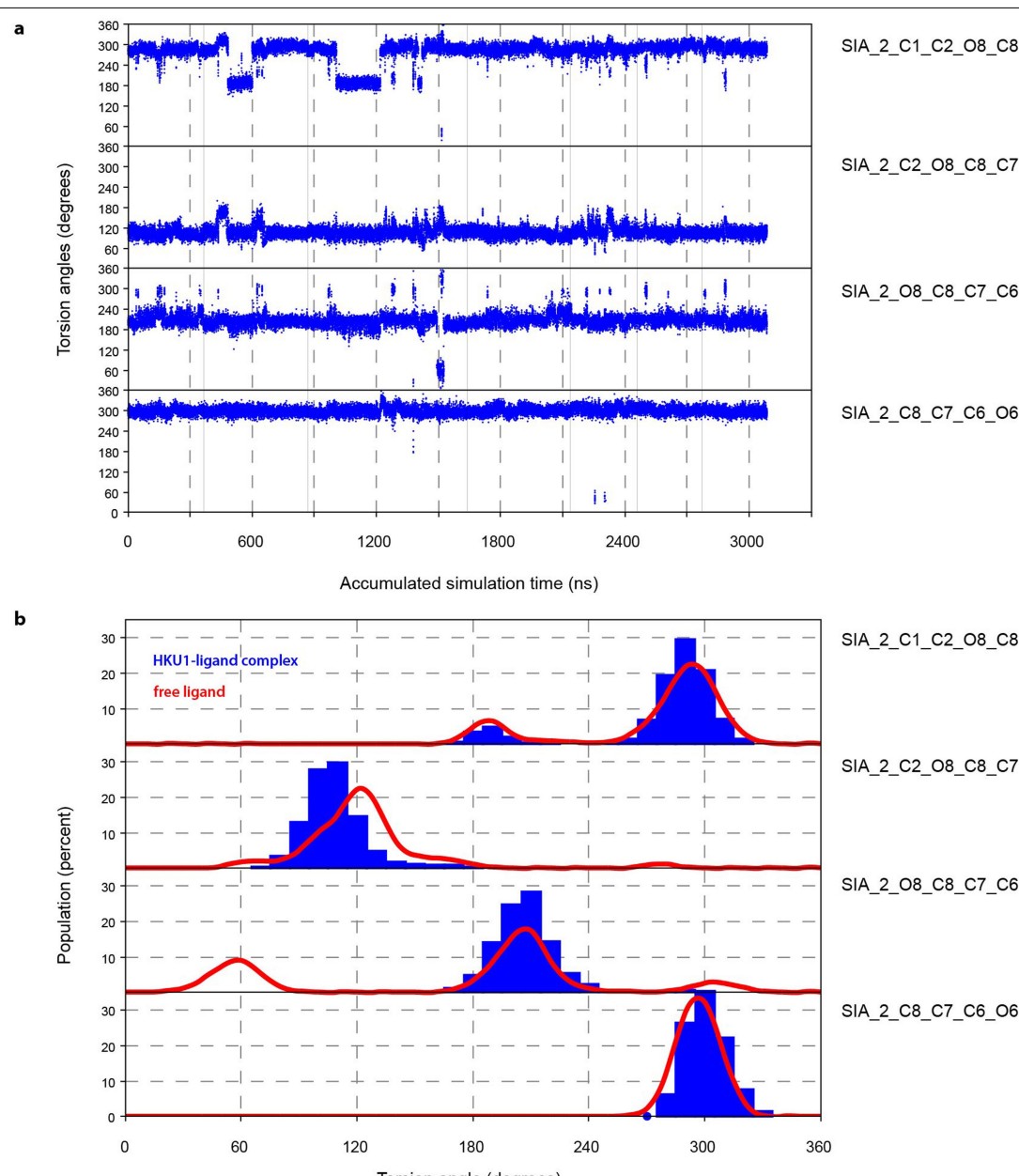

**Extended Data Fig. 9 | Dynamics of the α2-8 linkage of the disialoside in the complex.** Dynamics in the α2-8 linkage are reduced but remain possible when the disialoside is bound to HKU1-A Caen1. **a**, Trajectory plots of linkage torsions φ, ψ, γ and δ. In comparison to the dynamics of the disaccharide in the free state (Extended Data Fig. 7), there is a clear reduction in the conformational transition frequency for torsions φ and γ. **b**, Histograms of linkage torsions φ, ψ, γ and δ.

In comparison to the profiles of the disaccharide in the free state (red curves) there is a clear reduction in accessible conformational space for torsions γ. Whereas in the free state there are three population maxima, there is now a clear preference for a value around 210°. Data were derived from the same MD simulations shown in Extended Data Fig. 8.

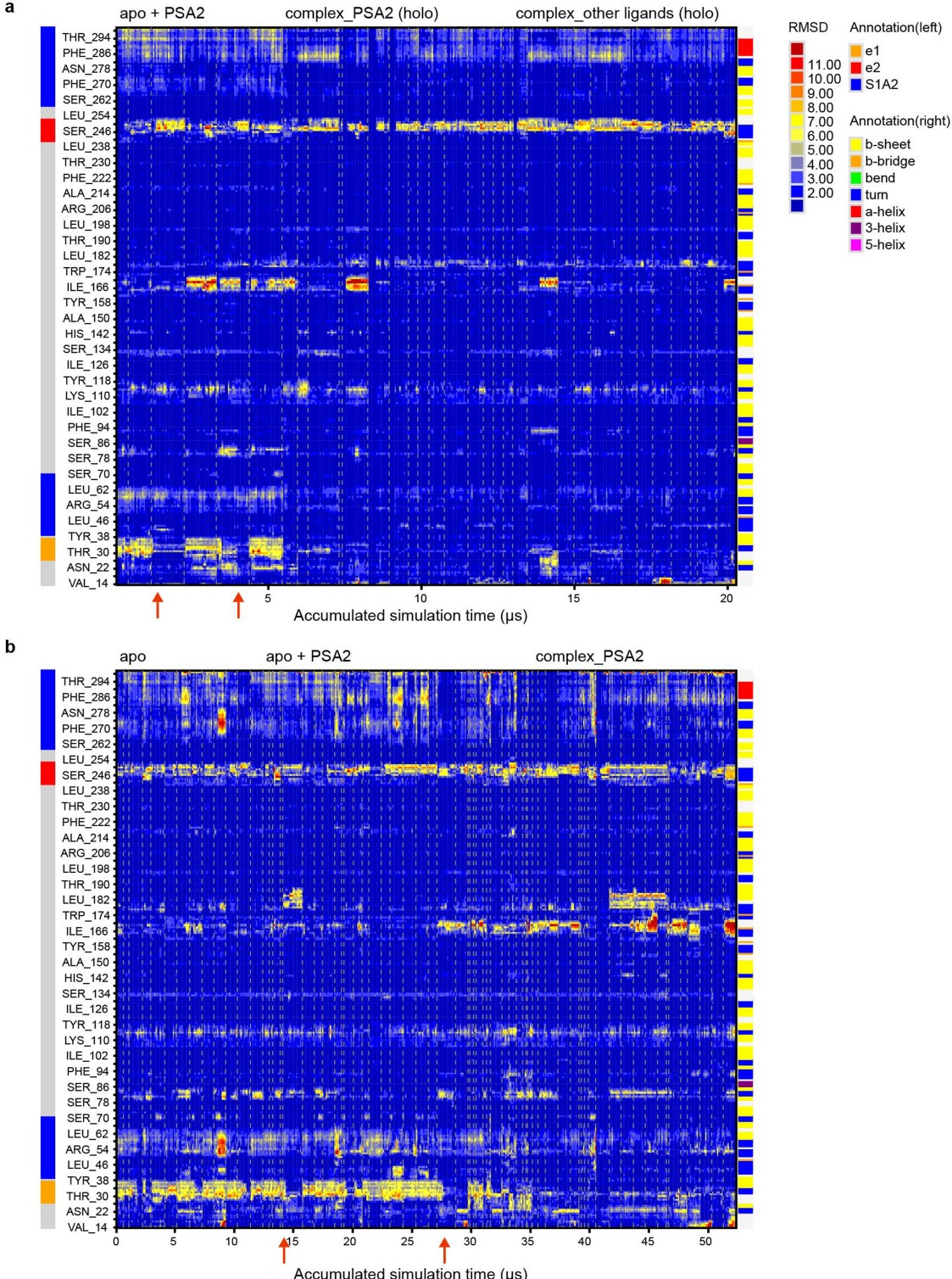

**Extended Data Fig. 10 | Dynamics of HKU1 spike S1^A.** All MD simulations performed with the Caen1 (**a**) and N1 sequence (**b**) were combined for an RMSD analysis based on the *holo* cryo-EM model as a reference structure. Conformational changes from the *apo* into the *holo* state are apparent as transitions from high (red) to low (blue) RMSD states. Simulations performed with the N1 strain sequence were based on the *apo* cryo-EM model of Caen1, replacing respective residues different in N1. Individual simulations (a: 33,

b: 69) are separated by vertical lines. The spontaneous conformational shifts of the e1 loop, observed in the simulations starting with the disialoside (PSA2) bound to the *apo* cryo-EM conformation of the Caen1 and N1 spike proteins are indicated with red arrows. Simulations based on the *holo* EM model with PSA2 or other Neu5,9Ac-containing ligands show ligand-induced stabilisation of the e1 conformational shift.

Dr. Raoul J. de Groot

# Reporting Summary

## Statistics

For all statistical analyses, confirm that the following items are present in the figure legend, table legend, main text, or Methods section.

| n/a | Confirmed | |
|---|---|---|
| ☐ | ☒ | The exact sample size (*n*) for each experimental group/condition, given as a discrete number and unit of measurement |
| ☒ | ☐ | A statement on whether measurements were taken from distinct samples or whether the same sample was measured repeatedly |
| ☒ | ☐ | The statistical test(s) used AND whether they are one- or two-sided<br>*Only common tests should be described solely by name; describe more complex techniques in the Methods section.* |
| ☒ | ☐ | A description of all covariates tested |
| ☒ | ☐ | A description of any assumptions or corrections, such as tests of normality and adjustment for multiple comparisons |
| ☐ | ☒ | A full description of the statistical parameters including central tendency (e.g. means) or other basic estimates (e.g. regression coefficient) AND variation (e.g. standard deviation) or associated estimates of uncertainty (e.g. confidence intervals) |
| ☒ | ☐ | For null hypothesis testing, the test statistic (e.g. $F$, $t$, $r$) with confidence intervals, effect sizes, degrees of freedom and $P$ value noted<br>*Give P values as exact values whenever suitable.* |
| ☒ | ☐ | For Bayesian analysis, information on the choice of priors and Markov chain Monte Carlo settings |
| ☒ | ☐ | For hierarchical and complex designs, identification of the appropriate level for tests and full reporting of outcomes |
| ☒ | ☐ | Estimates of effect sizes (e.g. Cohen's *d*, Pearson's *r*), indicating how they were calculated |

*Our web collection on statistics for biologists contains articles on many of the points above.*

## Software and code

Policy information about availability of computer code

| | |
|---|---|
| Data collection | Cryo-EM data was collected using EPU software (version 2.8.1). The MD simulations were performed using YASARA (version 22.9.24). |
| Data analysis | The cryo-EM maps were generated using MotionCor2, Relion (version 3.1.1), cryoSPARC live/cryoSPARC (version 4.1.0) and DeepEMhancer (version 0.13 - implemented through COSMIC2). Preliminary HKU1 model was generated using Phyre2 (version 2), and final models were built, refined and validated using UCSF Chimera (version 1.16), Coot (version 0.9.8.1), Elbow (version 1.20.1-4487), Phenix (version 1.19.2-4158) and molprobity (version 4.02-528). Analysis and visualisation was carried out with PDBePISA (version 1.52), Consurf (version 2.42), UCSF Chimera (version 1.16), UCSF ChimeraX (version 1.5), Clustal Omega (version 1.2.4) and CCP4 Superpose (version 1.0.0). Conformational Analysis Tools (CAT, www.md-simulations.de/CAT/ - version 2023) and VMD (version 1.9.2) were used for analysis and visualisation of MD trajectory data, general data processing and generation of scientific plots. |

For manuscripts utilizing custom algorithms or software that are central to the research but not yet described in published literature, software must be made available to editors and reviewers. We strongly encourage code deposition in a community repository (e.g. GitHub). See the Nature Portfolio guidelines for submitting code & software for further information.

## Data

Policy information about availability of data

All manuscripts must include a data availability statement. This statement should provide the following information, where applicable:
- Accession codes, unique identifiers, or web links for publicly available datasets
- A description of any restrictions on data availability
- For clinical datasets or third party data, please ensure that the statement adheres to our policy

The atomic models of the apo, holo, 1-up and 3-up HCoV-HKU1 spike have been deposited to the Protein Data Bank under the accession codes 8OHN, 8OPM, 8OPN and 8OPO. The globally and locally refined cryo-EM maps have been deposited to the Electron Microscopy Data Bank under the accession codes EMD-16882, EMD-17076, EMD-17077, EMD-17078, EMD-17079, EMD-17080, EMD-17081, EMD-17082 and EMD-17083. Data files pertaining to MD simulation results shown in Fig. 5b, and Extended Data Figs. 7-10 are made available via Zenodo under the DOI 10.5281/zenodo.7867090.

## Research involving human participants, their data, or biological material

Policy information about studies with human participants or human data. See also policy information about sex, gender (identity/presentation), and sexual orientation and race, ethnicity and racism.

| | |
|---|---|
| Reporting on sex and gender | N/A |
| Reporting on race, ethnicity, or other socially relevant groupings | N/A |
| Population characteristics | N/A |
| Recruitment | N/A |
| Ethics oversight | N/A |

Note that full information on the approval of the study protocol must also be provided in the manuscript.

# Field-specific reporting

Please select the one below that is the best fit for your research. If you are not sure, read the appropriate sections before making your selection.

☒ Life sciences ☐ Behavioural & social sciences ☐ Ecological, evolutionary & environmental sciences

For a reference copy of the document with all sections, see nature.com/documents/nr-reporting-summary-flat.pdf

# Life sciences study design

All studies must disclose on these points even when the disclosure is negative.

| | |
|---|---|
| Sample size | No statistical method was used to determine sample size. Sample sizes were determined by available electron microscopy time and the number of particles on electron microscopy grids. The sample size is sufficient to obtain structures at the reported resolution, as assessed by Fourier shell correlation. Three cryo-EM data sets were collected in total for this study. For the apo sample, 4207 movies were collected, 914772 particles were picked, 108396 particles went into the final global refinement and 325188 sub-particles went into the local refinement. For the holo sample, 4057 movies were collected, 956697 particles were picked. Of these, 44081 particles went into the global closed holo map, 36048 went into the 1-up map and 99174 particles went into the 3-up global structure. For the local refinements, 71458 sub-particles went into the closed holo map and 99174 sub-particles went into the local 3-up structure. For the mutant dataset, 896 movies were collected, 215843 particles were picked, 20452 went into the global refinement and 61356 sub-particles went into the local refinement.

The rationale behind the MD simulations was to simulate the conformational transitions between the two states observed by cryo-EM. In principle, it cannot be predicted beforehand how long this will take or whether it will be feasible at all with the computer resources available. The sample sizes (i.e. individual MD simulation times) for all investigated systems were chosen such that protein and ligand dynamics on the ns-$\mu$s time scale could be assessed with confidence. Accumulated simulation times are stated in the methods and range between 5 and 23 $\mu$s for the individual protein and protein-ligand systems studied. Individual simulations reached up to 1.6 $\mu$s. The total accumulated simulation time of 70 $\mu$s (for all systems) can be considered currently as very long. |
| Data exclusions | During cryo-EM image processing, micrographs with a CTF estimated resolution of worse than 10 Å were discarded and particles representing false picks or 'junk' particles were removed during 2D and 3D classification procedures. This is common practice for single-particle cryo-EM processing workflows. |
| Replication | The apo, holo and mutant datasets were collected in one session each and were not repeated. It is unattainable from time and cost to repeat cryo-EM data collection and processing on the exact same sample. For each reconstruction, two independent maps were refined in order to estimate resolution according to the recommended procedures in the field (the 'gold standard'). |

| | |
|---|---|
| | Replicas of MD simulations of individual protein-ligand simulation systems are detailed in the methods (6-34 per system) and individual results can be expected in Extended Data Fig. 10. The free disialoside ligand was simulated for 10x 1μs (Extended Data Fig. 7). |
| Randomization | During 3D refinement of the cryo-EM structures, particles were split into two half sets. For the MD experiments, randomization is only relevant during the initialization phase of an MD simulation where initial velocities are assigned to the individual atoms according to the Maxwell–Boltzmann distribution associated with the simulation temperature. |
| Blinding | This study does not involve any experiments where blinding would be applicable. |

# Behavioural & social sciences study design

All studies must disclose on these points even when the disclosure is negative.

| | |
|---|---|
| Study description | N/A |
| Research sample | N/A |
| Sampling strategy | N/A |
| Data collection | N/A |
| Timing | N/A |
| Data exclusions | N/A |
| Non-participation | N/A |
| Randomization | N/A |

# Ecological, evolutionary & environmental sciences study design

All studies must disclose on these points even when the disclosure is negative.

| | |
|---|---|
| Study description | N/A |
| Research sample | N/A |
| Sampling strategy | N/A |
| Data collection | N/A |
| Timing and spatial scale | N/A |
| Data exclusions | N/A |
| Reproducibility | N/A |
| Randomization | N/A |
| Blinding | N/A |

Did the study involve field work?  ☐ Yes  ☒ No

# Reporting for specific materials, systems and methods

We require information from authors about some types of materials, experimental systems and methods used in many studies. Here, indicate whether each material, system or method listed is relevant to your study. If you are not sure if a list item applies to your research, read the appropriate section before selecting a response.

## Materials & experimental systems

| n/a | Involved in the study |
|-----|----------------------|
| ☒ | ☐ Antibodies |
| ☐ | ☒ Eukaryotic cell lines |
| ☒ | ☐ Palaeontology and archaeology |
| ☒ | ☐ Animals and other organisms |
| ☒ | ☐ Clinical data |
| ☒ | ☐ Dual use research of concern |
| ☒ | ☐ Plants |

## Methods

| n/a | Involved in the study |
|-----|----------------------|
| ☒ | ☐ ChIP-seq |
| ☒ | ☐ Flow cytometry |
| ☒ | ☐ MRI-based neuroimaging |

# Eukaryotic cell lines

Policy information about cell lines and Sex and Gender in Research

| | |
|---|---|
| Cell line source(s) | ATCC HEK-293T |
| Authentication | Further authentication was not performed for this study. |
| Mycoplasma contamination | Mycoplasma testing was not performed for this study. |
| Commonly misidentified lines (See ICLAC register) | No commonly misidentified cell lines were used in this study. |

