## [Peer Review File · Nature]

Manuscript Title: Sialoglycan binding triggers spike opening in a human coronavirus

Reviewer Comments & Author Rebuttals

Reviewer Reports on the Initial Version:

Referees' comments:

Referee #1 (Remarks to the Author):

The manuscript from Pronker et al investigates a long-standing question in the coronavirus field: what is the trigger that causes the S1B domain (or RBD for many CoV spikes) to hinge from the 'down' conformation to the 'up' conformation, which is thought to be required for S1 shedding and S2 refolding to facilitate membrane fusion. Via cryo-EM studies on the HKU1-A spike, the authors demonstrate that binding of its disialoside receptor 9-O-Ac-Sia(α 2,8)Sia to the S1A domain (or NTD) induces local conformational changes that propagate all the way to the S1B domains, weakening interactions and allowing the S1B domains to hinge open. They observe both a '1-up' state and a '3-up' state, but surprisingly not a '2-up' state, which is apparently short-lived. The authors confirm that the conformational change is specific to the binding of the disialoside to the S1A domain by obtaining a structure of an HKU1-A spike with a W89A substitution that prevents disialoside binding. Molecular dynamics simulations provide additional support for the conformational dynamics observed in the EM studies.

These are important results for the field and the manuscript is very well written. The cryo-EM studies are performed well, and the authors take care not to over-interpret their data given the resolutions obtained. The 3D variability analysis on the apo and 1-up datasets demonstrates that their method for particle classification is sufficient to discriminate between open and closed S1B domains, and thus that there is not an appreciable fraction (if any) open particles in their apo dataset. I am not an expert on the MD simulations, so another reviewer will need to comment on those studies. If there is room, the authors may want to comment on the differences in the particle distributions observed in their dataset between closed, 1-up, 2-up, and 3-up, and those of datasets from SARS, MERS, and SARS2 spikes. My only other comment is that in Extended Data Table 2, there are too many significant figures for some of the values. B-factors should not be listed to the hundredths of an Å², nor should angle deviations be listed to the thousandths of angle. Once corrected, I recommend publication in Nature, and I expect this manuscript to be well received by the coronavirus field and general readers.

Referee #2 (Remarks to the Author):

This manuscript reports cryo-EM structures of CoV HKU1-A spikes, where addition of receptor analog 9-O-Ac-Sia apparently stimulates 1-up and 3-up opening transitions, noted in ~75% of samples. These transitions were not observed in the absence of the receptor or with a sialoglycan-binding defective mutant, supporting the idea that binding of the sialoglycan primary receptor in the S1A site then leads to a conformational transition that exposes the S1B site for secondary receptor binding. The manuscript is clearly written. The findings of the work are novel and of key significance, providing a new mechanistic understanding of the balance between cell adhesion and immune evasion in CoVs.

Molecular dynamics simulations are used to 1) map regions of the intact spike protein that are shielded by glycans, 2) identify disialoside conformers that allowed modeling of the reducing-end residue into the cryo-EM complex, and 3) support the sialoglycan-induced conformational

transition via simulations of the S1A domain.

The data from (1) is presented in an image in a supplemental figure, but not interpreted or discussed. The simulations used for (2) are not described in Methods, so it is not clear what force field or software were used, or what the simulation conditions were; thus, all information required for review or reproducibility is absent, and should be added to the manuscript during revisions.

The simulations for (1) and (3) are described in Methods, but most key information that would allow thorough review or reproducibility of the protocol is likewise absent. Simulations were performed in the YASARA engine, applying the "AMBER14" force field. Since AMBER entails a large family of force fields, there is some ambiguity as to which parameter sets were actually applied. The references listed for AMBER14 (54-56) are the ff14SB paper (which makes sense for the protein portion of the system), but then also the GLYCAM06 paper and a paper for a new charge model for GAFF2, which is a general force field with parameter coverage for the functional groups of many small molecules. GLYCAM06 (a widely used and trusted carbohydrate force field) is hopefully what was applied to the glycan portion of the system, as applying GAFF2 would surely lead to untrustworthy dynamics. It is unclear to me why the paper for a new GAFF2 charge model is cited here; nothing in the manuscript indicates its relevance to the study. Best to clarify which force fields were used in the manuscript text, and also indicate which water model was used for the explicit solvent (perhaps TIP3P). Exactly which GLYCAM06 version was used should be specified, as this is not apparent from the citation (e.g., GLYCAM_06j-1).

A current issue with running GLYCAM06 in MD engines other than AMBER's sander/pmemd/pmemd.cuda is that GLYCAM06 uses a different 1-4 electrostatic and nonbonded scaling factor than AMBER protein force fields. While AMBER MD engines handle this, NAMD does not, and GROMACS requires a 3rd party conversion software that correctly treats negative dihedral force constants. The issue is that NAMD and GROMACS, for example, require the 1-4 scaling factors to be set globally. I was not able to find whether YASARA handles mixed 1-4 scaling factors. If it does not, and 1-4 scaling factors are set globally, then either the protein or the glycan portion of the simulation systems may exhibit some incorrect dynamics, depending on which has the incorrect 1-4 scaling factors.

Surely simulation systems were minimized prior to performing molecular dynamics, but the algorithm and protocol for doing this is not described. Were the systems heated or simply thermalized? For how long was the system equilibrated before production molecular dynamics? What were the criteria for determining the systems had reached equilibrium?

Beyond periodic boundaries, temperature 310 K, rescaling the simulation box size to maintain correct water density, and using a 4-fs timestep, none of the common descriptions of molecular dynamics simulation conditions were reported. The thermostat algorithm for constant temperature, and how often it is invoked is not indicated. If the thermostat algorithm includes a stochastic contribution, is each simulation in the accumulated sampling using a unique random number seed? Box rescaling is essentially maintaining a constant pressure, but any more specific barostat algorithm, or how often rescaling was performed is not indicated. Was scaling isotropic? To be clear in the text, constant temperature, pressure, and particle number is commonly called isothermal-isobaric conditions or NPT. What long-range cutoffs for electrostatic and van der Waals interactions were used? How were long-range electrostatics computed? Were long-range electrostatics computed every timestep or on multiples? Coming back to the simulation systems, are the boxes cubic, orthorhombic, truncated octahedral, etc. and what is the buffer of water between solute and box edge? These are examples of the details that would have enabled more thorough review of the protocol's validity, and should be added to ensure reproducibility in the final version of the manuscript.

Also, the use of a 4-fs timestep in stable simulations absolutely requires constraints on the dynamics, but these are not indicated. Stretching and compressing of bonds to hydrogen are

commonly constrained, but 4-fs requires either additional angle constraints or schemes like hydrogen mass repartitioning. Which is it here? To get to timesteps this large, you must give something up in the dynamics, so this factor should be noted.

Finally, the MD simulation data used to support the main conclusions of the manuscript, that binding of the sialoglycan receptor stimulates a conformational change, were carried out on simulations of the S1A domain alone. Since this system was heavily truncated, were some restraints applied to any part of it to prevent unfolding (e.g., harmonic positional restraints)? If so, the restraint scheme should be reported. If the whole S1A domain is completely unrestrained, how can the authors be sure that the conformational changes they observe in the domain are not the result of the system relaxing outside of the context of the rest of the spike structure? Because the disialoside receptor analog has only two residues but the native glycan the spike would bind is larger, can the authors comment on any limitations of the simulations; might the extended glycan structure further impact observed dynamics?

In general, I found the simulation results compelling and supportive of the conclusions of the work, however, I would ask that these concerns and omissions be addressed before publication.

Author Rebuttals to Initial Comments:

Reply to the Referees

Referee #1 (Remarks to the Author):

The manuscript from Pronker et al investigates a long-standing question in the coronavirus field: what is the trigger that causes the S1B domain (or RBD for many CoV spikes) to hinge from the 'down' conformation to the 'up' conformation, which is thought to be required for S1 shedding and S2 refolding to facilitate membrane fusion. Via cryo-EM studies on the HKU1-A spike, the authors demonstrate that binding of its disialoside receptor 9-O-Ac-Sia(α 2,8)Sia to the S1A domain (or NTD) induces local conformational changes that propagate all the way to the S1B domains, weakening interactions and allowing the S1B domains to hinge open. They observe both a '1-up' state and a '3-up' state, but surprisingly not a '2-up' state, which is apparently short-lived. The authors confirm that the conformational change is specific to the binding of the disialoside to the S1A domain by obtaining a structure of an HKU1-A spike with a W89A substitution that prevents disialoside binding. Molecular dynamics simulations provide additional support for the conformational dynamics observed in the EM studies.

These are important results for the field and the manuscript is very well written. The cryo-EM studies are performed well, and the authors take care not to over-interpret their data given the resolutions obtained. The 3D variability analysis on the apo and 1-up datasets demonstrates that their method for particle classification is sufficient to discriminate between open and closed S1B domains, and thus that there is not an appreciable fraction (if any) open particles in their apo dataset. I am not an expert on the MD simulations, so another reviewer will need to comment on those studies.

We thank the Referee for their positive assessment of our manuscript. Our responses to the points raised are included below.

If there is room, the authors may want to comment on the differences in the particle distributions observed in their dataset between closed, 1-up, 2-up, and 3-up, and those of datasets from SARS, MERS, and SARS2 spikes.

We considered commenting on differences in particle distribution for the viruses mentioned. For SARS-CoV and MERS-CoV this would be possible [1,2]. However, in the case of SARS-CoV-2, we found particle distributions to vary across studies [3,4], and between different variants [5,6], making it difficult to provide and discuss numbers concisely without breaking the flow of the text. Related to this, for SARS-CoV-2, we noted a trend developing over the course of the pandemic. In an earlier version of our manuscript, we contemplated including an additional discussion about the current evolutionary trajectory of the SARS-CoV-2 spike towards a more closed state (please see below). Eventually, we decided against it as we felt it might be too speculative and too much off topic in a manuscript focusing on HKU1. Also, the manuscript in its current form is close to the maximum word count allowed.

[In this view, sarbeco- and merbecoviruses, spontaneously exposing S1^B, would not be exceptions but part of a mechanistic spectrum with other CoVs relying on specific triggers such as binding to primary receptors via S1^A. In natural reservoir host populations, the unyielding pressure of humoral immunity might select for cued mechanisms for spike opening, in order to conceal neutralising epitopes. Conversely, during virgin soil epidemics/epizootics, i.e. when CoVs colonise immunologically naive populations, dynamic sampling of the open spike conformation may confer a fitness advantage. In line with this hypothesis, SARS-CoV-2 at the very onset of the pandemic acquired the D614G mutation, allowing the spikes to more readily assume S1^B-up conformations, while those of recent Omicron variants predominantly adopt the closed state.]

References:

[1] Gui et al. [doi: 10.1038/cr.2016.152] - In this study, 28% of SARS-CoV spikes were closed, and the rest were open/partially open.

[2] Yuan et al. [doi: 10.1038/ncomms15092] - MERS-CoV spikes were observed in two states, with one (40% of particles) or two (60% of particles) S1^B domains in the open state.

[2] Wrapp et al. [doi: 10.1126/science.abb2507] - Here, 35% of SARS-CoV-2 spikes had one open S1^B and the rest were closed.

[3] Walls et al. [doi: 10.1016/j.cell.2020.02.058] - In contrast to reference 2, 53% of spikes had one open S1^B and the rest were closed.

[4] Díaz-Salinas et al. [doi: 10.1101/2021.10.29.466470] - Using FRET experiments, the authors observed 35% open spike for the Wuhan isolate and 59% for D614G.

[5] Gobeil et al. [doi: 10.1126/science.abi6226] - The authors observed a higher proportion of open spikes for the B.1.1.7 spikes relative to the D614G variant.

[6] Calvaresi et al. [doi: 10.1038/s41467-023-36745-0] - Using HDX-MS, the authors show that the alpha, beta and delta spikes are mostly open, and omicron predominantly adopts the closed conformation, presumably to evade humoral immunity.

My only other comment is that in Extended Data Table 2, there are too many significant figures for some of the values. B-factors should not be listed to the hundredths of an Å², nor should angle deviations be listed to the thousandths of angle.

Point well taken. This has now been corrected (see supplementary table 2).

Once corrected, I recommend publication in Nature, and I expect this manuscript to be well received by the coronavirus field and general readers.

We hope that the above responses have addressed the Referee's concerns.

Referee #2 (Remarks to the Author):

This manuscript reports cryo-EM structures of CoV HKU1-A spikes, where addition of receptor analog 9-O-Ac-Sia apparently stimulates 1-up and 3-up opening transitions, noted in ~75% of samples. These transitions were not observed in the absence of the receptor or with a sialoglycan-binding defective mutant, supporting the idea that binding of the sialoglycan primary receptor in the S1A site then leads to a conformational transition that exposes the S1B site for secondary receptor binding. The manuscript is clearly written. The findings of the work are novel and of key significance, providing a new mechanistic understanding of the balance between cell adhesion and immune evasion in CoVs.

Molecular dynamics simulations are used to 1) map regions of the intact spike protein that are shielded by glycans, 2) identify disialoside conformers that allowed modeling of the reducing-end residue into the cryo-EM complex, and 3) support the sialoglycan-induced conformational transition via simulations of the S1A domain.

We thank the Referee for insightful comments and constructive criticism regarding the MD analysis. We acknowledge that the information provided in the original submission may have been too sparse to judge the quality of our simulations. The original MD Methods section was written with the aim of keeping it general and concise. We used YASARA, which is a commercial software package developed and distributed by YASARA Biosciences GmbH (<http://www.yasara.org>) to set up and perform the MD simulations. YASARA provides automated workflows for (standard) production MD simulations (with many default settings) and therefore works slightly differently than MD codes distributed by academic groups such as GROMACS, AMBER or NAMD, where a significant number of parameters need to be adjusted in the setup before a production simulation can be performed. However, we concur that key information was missing and have significantly extended the MD methods section according to the recommendations of the Referee. The newly added Methods text are indicated in blue. The individual points raised by the Referee are addressed in detail below.

The data from (1) is presented in an image in a supplemental figure, but not interpreted or discussed.

Point well taken. We have altered the main text (l. 79-83) to read: *'Using site-specific glycosylation patterns of HKU1-B³², we performed molecular dynamics (MD) simulations of the fully glycosylated S ectodomain trimer. HKU1-A S is largely shielded by glycans leaving only a few regions exposed, most notably the sialic acid-binding site in domain S1^A (Extended Data Fig. 2b).'* We have also included the image of the glycosylated S ectodomain to Extended Data Fig. 2, so that it would appear alongside the main text figure.

The simulations used for (2) are not described in Methods, so it is not clear what force field or software were used, or what the simulation conditions were; thus, all information required for review or reproducibility is absent, and should be added to the manuscript during revisions.

The missing information is added to the Methods section (please see below).

The simulations for (1) and (3) are described in Methods, but most key information that would allow thorough review or reproducibility of the protocol is likewise absent.

The missing key information is added to the Methods section (please see below).

Simulations were performed in the YASARA engine, applying the "AMBER14" force field. Since AMBER entails a large family of force fields, there is some ambiguity as to which parameter sets were actually applied. The references listed for AMBER14 (54-56) are the ff14SB paper (which makes sense for the protein portion of the system), but then also the GLYCAM06 paper and a paper for a new charge model

for GAFF2, which is a general force field with parameter coverage for the functional groups of many small molecules. GLYCAM06 (a widely used and trusted carbohydrate force field) is hopefully what was applied to the glycan portion of the system, as applying GAFF2 would surely lead to untrustworthy dynamics. It is unclear to me why the paper for a new GAFF2 charge model is cited here; nothing in the manuscript indicates its relevance to the study. Best to clarify which force fields were used in the manuscript text, and also indicate which water model was used for the explicit solvent (perhaps TIP3P). Exactly which GLYCAM06 version was used should be specified, as this is not apparent from the citation (e.g., GLYCAM_06j-1).

AMBER14 includes the most recent widely used AMBER force field (also known as AMBER14SB or ff14sb). In addition, YASARA will assign parameters from the GLYCAM06j and Lipid17 force fields if they contain a matching residue. If not, it will derive GAFF2/AM1BCC parameters otherwise (AUTOSMILES method). Parameters for ions come from extensive optimization studies (all references given below). In our simulations, GLYCAM06j parameters are used for the N-glycans and the disialoside ligand. GAFF2 parameters were not used, therefore this reference has been deleted from the manuscript.

Development and Testing of a General AMBER Force Field. Wang J, Wolf RM, Caldwell JW, Kollman PA and Case DA (2004) *J.Comput.Chem.* 25, 1157-1174 PMID15116359

Fast, efficient generation of high-quality atomic charges. AM1-BCC model: II. Parameterization and validation. Jakalian A, Jack DB and Bayly CI (2002) *J.Comput.Chem.* 23, 1623-1641 PMID12395429

Lipid14: The Amber Lipid Force Field. Dickson C, Madej B, Skjevik A, Betz R, Teigen K, Gould I, Walker R (2014) *J.Chem.Theory Comput.* 10, 865-879 PMID24803855

GLYCAM06: A generalizable biomolecular force field for carbohydrates. Kirschner K, Yongye A, Tschampel S, Gonzalez-Outeirino J, Daniels C, Foley B, Woods R (2008) *J.Comput.Chem.* 29, 622-655 PMID17849372

monovalent ions:

Systematic parameterization of monovalent ions employing the nonbonded model. Li P, Song LF, Merz Jr KM (2015) *J.Chem.Theory Comput.* 11, 1645-1657 PMID26574374

divalent ions:

Rational design of particle mesh Ewald compatible Lennard-Jones parameters for +2 metal cations in explicit solvent. Li P, Roberts BP, Chakravorty DK, Merz Jr KM (2013) *J.Chem.Theory Comput.* 9, 2733-2748 PMID23914143

ions with higher valency:

Parameterization of highly charged metal ions using the 12-6-4 LJ type nonbonded model in explicit water. Li P, Song LF, Merz Jr KM (2015) *J.Phys.Chem.B* 119, 883-895 PMID25145273

sugars with sulfate groups:

Force field parameters for sulfates and sulfamates based on ab initio calculations: extensions of AMBER and CHARMM force fields. Huige CJM, Altona C (1995) *J.Comp.Chem.* 16, 56-79

A current issue with running GLYCAM06 in MD engines other than AMBER's sander/pmemd/pmemd.cuda is that GLYCAM06 uses a different 1-4 electrostatic and nonbonded scaling factor than AMBER protein force fields. While AMBER MD engines handle this, NAMD does not, and GROMACS requires a 3rd party conversion software that correctly treats negative dihedral force constants. The issue is that NAMD and GROMACS, for example, require the 1-4 scaling factors to be set globally. I was not able to find whether YASARA handles mixed 1-4 scaling factors. If it does not, and 1-4 scaling factors are set globally, then either the protein or the glycan portion of the simulation systems may exhibit some incorrect dynamics, depending on which has the incorrect 1-4 scaling factors.

YASARA uses mixed 1-4 scaling factors in AMBER14: SCEE=1.2 and SCNB=2.0 for ff14sb residues and SCEE=1.0 and SCNB=1.0 for GLYCAM06 residues.

Surely simulation systems were minimized prior to performing molecular dynamics, but the algorithm and protocol for doing this is not described.

The "energy minimization" macro in YASARA has been used, which does the following: after removal of conformational stress by a short steepest descent minimization, the procedure continues with simulated annealing (timestep 2 fs, atom velocities scaled down by 0.9 every 10th step) until convergence was reached, i.e. the energy improved by less than 0.05 kJ/mol per atom during 200 steps.

Were the systems heated or simply thermalized?

Since we applied position restraints on the protein heavy atoms during the first equilibration phase of the MD, we simply used the default temperature control scheme of the YASARA MD macro (command 'TempCtrl Rescale'), which keeps the time average macroscopic temperature at the requested value by rescaling the atom velocities using a weak coupling thermostat. To minimize the influence of rescaling, YASARA does not use the strongly fluctuating instantaneous microscopic temperature to rescale velocities at each simulation step (classic Berendsen thermostat). Instead, the scaling factor is calculated according to Berendsen's formula from the time average temperature. This ensures that the thermostat does not inhibit the continuous interconversion between kinetic and potential energy by enforcing a fixed temperature and thus kinetic energy, and avoids artifacts often observed for the Berendsen thermostat. If constraints are present, YASARA controls the temperature for solute and water independently, since the use of different constraint algorithms, e.g. LINCS for the solute and SETTLE for the water, could otherwise lead to significantly different temperatures in the cell.

For how long was the system equilibrated before production molecular dynamics?

What were the criteria for determining the systems had reached equilibrium?

The solvent was equilibrated for at least 3 ns with position restraints on the protein (first all heavy atoms, then only backbone atoms). Finally, the restraints were removed with the exception of those that are used to maintain the orientation of the system in the cuboid box. When the RMSD of the protein reached a plateau at about 2 Å, this was considered as an 'equilibrated system'.

Beyond periodic boundaries, temperature 310 K, rescaling the simulation box size to maintain correct water density, and using a 4-fs timestep, none of the common descriptions of molecular dynamics simulation conditions were reported. The thermostat algorithm for constant temperature, and how often it is invoked is not indicated. If the thermostat algorithm includes a stochastic contribution, is each simulation in the accumulated sampling using a unique random number seed? Box rescaling is essentially maintaining a constant pressure, but any more specific barostat algorithm, or how often rescaling was performed is not indicated. Was scaling isotropic? To be clear in the text, constant temperature, pressure, and particle number is commonly called isothermal-isobaric conditions or NPT. What long-range cutoffs for electrostatic and van der Waals interactions were used? How were long-range electrostatics

computed? Were long-range electrostatics computed every timestep or on multiples? Coming back to the simulation systems, are the boxes cubic, orthorhombic, truncated octahedral, etc. and what is the buffer of water between solute and box edge? These are examples of the details that would have enabled more thorough review of the protocol's validity, and should be added to ensure reproducibility in the final version of the manuscript.

This information is now given in the revised Methods or available from references cited that describe the background of the methods implemented and used within YASARA.

Also, the use of a 4-fs timestep in stable simulations absolutely requires constraints on the dynamics, but these are not indicated. Stretching and compressing of bonds to hydrogen are commonly constrained, but 4-fs requires either additional angle constraints or schemes like hydrogen mass repartitioning. Which is it here? To get to timesteps this large, you must give something up in the dynamics, so this factor should be noted.

To allow larger timesteps, bonds to hydrogens and certain bond angles involving hydrogens are constrained. The YASARA documentation recommends a 4 fs timestep for structures with severe errors or higher temperatures, but 5 fs timesteps do also produce stable trajectories for pre-equilibrated systems based on our experience. The technical background of using larger timesteps in YASARA MD simulation is extensively discussed in the reference given:

Krieger E, Vriend G (2015) New ways to boost molecular dynamics simulations. *J Comput Chem* 36:996–1007. doi: 10.1002/jcc.23899

Finally, the MD simulation data used to support the main conclusions of the manuscript, that binding of the sialoglycan receptor stimulates a conformational change, were carried out on simulations of the S1A domain alone. Since this system was heavily truncated, were some restraints applied to any part of it to prevent unfolding (e.g., harmonic positional restraints)? If so, the restraint scheme should be reported.

Unfortunately, it was not possible for us to simulate the complete ectodomain over as many μ s as we have done for the S1^A system. For our analysis of the isolated domain, harmonic position restraints were used constantly throughout the simulations on selected atoms. To increase productivity, we simulated the systems in rectangular cuboid simulation boxes. Therefore, we needed to make sure that no overall rotation of the system occurred. We selected atoms that formed a pseudo-sub-domain remote from the binding site for the restraints. For S1^A, they extended to residue 299 (the S1^A C-terminus), which solved the issue of the potential artificial unfolding of the truncated S1A domain. We have now added the following text to the Methods section: *'Harmonic position restraints (stretching force constant = 1 N/m) were applied to protein backbone atoms of residues 48-65 and 264-299 of the S1^A system to prevent system rotation in the cuboid box and to deal with the 'artificially loose end' at residue 299. The average RMSD of the protein C-alpha atoms was monitored to check the overall stability of the simulation.'*

If the whole S1A domain is completely unrestrained, how can the authors be sure that the conformational changes they observe in the domain are not the result of the system relaxing outside of the context of the rest of the spike structure?

Restraints were applied as explained above.

Because the disialoside receptor analog has only two residues but the native glycan the spike would bind is larger, can the authors comment on any limitations of the simulations; might the extended glycan structure further impact observed dynamics?

This is an interesting question, but not within the scope of the present study. The research question behind the MD simulations was to explore whether the conformational changes in S1^A, in particular,

changes in e1 topology as observed by cryo-EM, could be corroborated independently. Thus, the MD simulations were designed to reproduce binding events *in silico* under identical conditions, which would entail using a disialoside as a receptor ligand. The transition of the binding site into a stable post-binding state occurs on a similar time scale as that of our individual simulations (up to 1.6 μ s) but with the odds of occurrence during each measurement <1 . Hence, multiple simulations were done per condition with the transition occurring frequently enough to describe the details of how such a transition may occur in reality. However, the data do not allow solid conclusions on the macroscopic kinetics of the transition between the two states. Without such robust *statistical* data, answering a question on the impact of a modification of the system is difficult and would require excessive computer time to allow for a quantitative, i.e. statistically significant comparison of dynamics with different ligands. Of note, our simulations of the S1^A domain system, as described in the present manuscript, are already quite extensive. Testing different conditions and controls entailed 92 simulations, 72 μ s in total, and took almost 500 days to compute. On a final note, based on the stable conformation of the disialoside as observed both by cryo-EM and MD, the extended glycan chain of natural glycans would project away from S1^A into the solvent and thus appears unlikely to impact e1 dynamics directly.

In general, I found the simulation results compelling and supportive of the conclusions of the work, however, I would ask that these concerns and omissions be addressed before publication. We hope that the concerns and omissions of the Referee have been adequately addressed.

Reviewer Reports on the First Revision:

Referees' comments:

Referee #1 (Remarks to the Author):

The reviewers have adequately addressed my prior comments in the revised manuscript.

Referee #2 (Remarks to the Author):

The revised version of the manuscript has addressed my concerns. The MD portion of the manuscript has been carried out carefully and the results are robust. Sufficient details of the MD protocol are now included to allow readers to follow what was done and have high confidence in the outcome. It is very satisfying to see MD show the local conformational changes induced by sialoglycan binding, the end states of which are observed by EM. The combination of techniques provides important insights into the role of the primary receptor and mechanism of secondary RBD exposure. The findings are of broad significance to the study of coronaviruses. I fully support publication of the manuscript in its revised form.

Jodi A. Hadden-Perilla